

**Modelling of long term Zn, Cu, Cd, Pb dynamics from soils fertilized with organic**
**amendments.**
Claudia Cagnarini[1,2], Stephen Lofts[3], Luigi Paolo D'Acqui[4], Jochen Mayer[5], Roman Grüter[6],
Susan Tandy[6,8], Rainer Schulin[6], Benjamin Costerousse[7], Simone Orlandini[1], Giancarlo Renella[9]
1, Dipartimento di Scienze e Tecnologie Agrarie, Alimentari, Ambientali e Forestali (DAGRI),
Piazzale delle Cascine, 18, 50144 Firenze, Italy
2, UK Centre for Ecology & Hydrology, Environment Centre Wales, Deiniol Road, Bangor,
Gwynedd, LL57 2UW, United Kingdom
3, UK Centre for Ecology & Hydrology, Lancaster Environment Centre, Library Avenue, Bailrigg,
Lancaster, LA1 4AP, United Kingdom
4, Istituto di Ricerca sugli Ecosistemi Terrestri, CNR, Area della Ricerca di Firenze, Via Madonna
del Piano 10, 50019 Sesto Fiorentino (Firenze), Italy
5, Agroscope, Dept Agroecology and Environment, Reckenholzstrasse 191, 8046 Zurich,
Switzerland
6, Institute of Terrestrial Ecosystems, ETH Zurich, Universitätstrasse 16, 8092 Zurich, Switzerland
7, Institute of Agricultural Sciences, ETH, Universitätstrasse 2, 8092 Zurich, Switzerland



8, Rothamsted Research, North Wyke, Okehampton, Devon, EX20 2SB, United Kingdom
9, Department of Agronomy, Food, Natural Resources, Animals and Environment (DAFNAE),
viale dell'Università 16, 35020 Legnaro (Pd), Italy
Corresponding author: Claudia Cagnarini, Email: ccagnar@ceh.ac.uk
Key words: trace elements, farmyard manure, sewage sludge, compost, sustainability, Switzerland
Declaration of interest: none.



**Abstract**
Soil contamination by trace elements (TEs) is a major concern for sustainable land management.
One potential source of excessive inputs of TEs into agricultural soils are organic amendments.
Here, we use dynamic simulations carried out with the IDMM-ag model to describe observed
trends of topsoil Zn, Cu, Pb and Cd concentrations in a long-term crop trial in Switzerland, where
soils plots have been treated with differing organic amendments, particularly farmyard manure,
sewage sludge and compost. IDMM-ag requires the definition of a parsimonious set of boundary
conditions.
The model adequately reproduced the metal EDTA-extractable concentrations in ZOFE when site-
specific soil lateral mixing, due to mechanically ploughing of small plots, was introduced.
Calibration of an additional metal input flux was necessary to fit the measured data, indicating that
knowledge gaps in quantifying historical metal inputs can affect field-scale simulations even in a
well-characterized field. Projections of soil metal content in the long-term showed that, under
stable organic amendment application rates, Zn and Cu labile concentrations might pose
toxicological hazard for the soil ecosystem, particularly in the sewage sludge-amended plots. The
sewage sludge topsoil was characterized by some variability in the organic matter composition,
potentially due to the applied sewage sludge quality, which might affect the metal lability: this
effect should be accounted for in models
This study takes a step forward in assessing potential and limitations of the IDMM-ag model to
predict TEs long-term dynamics in agricultural fields, paving the way to quantitative applications
of TEs modelling at field and larger scales.



## 1. Introduction

Trace elements (TEs) are naturally present in soils due to mineral weathering and biogeochemical cycles. Many TEs, particularly cationic metals, are persistent in many topsoils, but can also leach to surface waters, with possible toxicological effects on the whole ecosystem. Several TEs such as zinc (Zn), copper (Cu) and nickel (Ni) play important roles in biochemical processes and are therefore essential for living organisms at low concentrations, though they can become toxic to biota at high concentrations; therefore, their presence in soil can be tolerable in a relatively narrow range of values (Adriano, 2005). In contrast, other TEs like lead (Pb) and cadmium (Cd), which are not physiologically active, are generally toxic to living organisms at low concentrations, and their accumulation in soil is of particular concern. Excessive uptake of trace elements by crop plants and their enrichment in edible parts can pose significant risks to human health by entering into the food chain (Mcgrath & Zhao, 2015). Accumulation of TEs in cultivated soils is widespread and is mainly caused by application of low grade agrochemicals, organic fertilizers and sewage sludge (Toth, Hermann, Da Silva, & Montanarella, 2016). In a European Union wide survey, Ballabio et al. (2018) reported that agricultural soils represent the environmental matrix with a high enrichment potential in TEs, and for example land cover and management are better predictors of soil Cu concentrations than natural soil formation factors. Due to limited natural availability of nutrient elements such as phosphorus (P), which is extracted from phosphate rocks (Roberts, 2014), and high energy consumption for the industrial production of mineral nitrogen (N) fertilizers, the application of organic amendments is considered a more sustainable option for agricultural soil management (Diacono & Montemurro, 2010). Organic amendments can have the additional benefit of increasing the soil organic matter (SOM) content, which usually enhances soil fertility and contributes to carbon sequestration from the atmosphere (Smith, 2016). However,



the application of such amendments, such as farmyard manure, compost and digestates of bio-
wastes, can also introduce TEs into agricultural soils. Application of sewage sludge into
agricultural soils can be even more problematic, as sewage sludge often contains concentrations
of various TEs such as Cd, Cu, Pb, and Zn up to 30 times their  concentrations in soil (Hudcova,
Vymazal, and Rozkosny, 2019; JRC, 2012; NEBRA, 2015).
Once in the soil, the fate of the TEs is controlled by multiple soil properties, as determined by land
use history, climatic forcing and geological setting. The soil pH,  soil and dissolved organic matter
(SOM, DOM) contents, the quantity and chemical composition of reactive minerals such as clay
minerals and metal (oxy)hydroxides, are all known to influence the speciation and solubility of
TEs in soils (Gu & Evans, 2008). Furthermore, the TEs speciation can influence the toxicological
hazard of the TEs, particularly to organisms that are directly exposed to soils, such as plants and
earthworms. In the context of long-term TEs accumulation due to regular organic amendment or
other additions, predicting the long-term speciation and dynamics of TEs is useful to support
decisions on ecosystem management and human health protection. Overall, owing to the TEs
reaction with the soil solid phases, repeated application of organic amendments can increase their
concentrations in agricultural soils through direct reactions or their physical entrapment into the
organo-mineral aggregates. This can lead to exceedance of their concentrations of the
environmental legislation thresholds.
Dynamic models are useful in predicting the accumulation, bioavailability and potential uptake of
TEs in soils, particularly for the projection of future trends. Reliable models can critically support
land use and land management decisions, and regulatory initiatives. Models for TEs dynamics
exist at a number of levels of complexity, from those with a site–specific, mechanistic approach
requiring highly detailed input information and calibration (Bonten, Groenenberg, Meesenburg, &



de Vries, 2011), to relative simple mass balance approaches applicable at large scale (Six &
Smolders, 2014). Models have been used to simulate behaviour and uptake of TEs at specific
agricultural sites subject to metal inputs (Bergkvist & Jarvis, 2004; Ingwersen & Streck, 2006),
based on site–specific calibration, but such models are not readily applicable at large scale without
generalisation of the parameterisation as a function of local climatic, hydrological and soil
physico-chemical properties.
Among models for determining the TE dynamics in soil, The IDMM (Intermediate Dynamic
Model for Metals) (Lofts, Tipping, Lawlor, & Shotbolt, 2013; Xu, Lofts, & Lu, 2016) is an
example of a dynamic model which allows general application, given a reasonably parsimonious
set of input data. It is intended for long term application from decades to centuries and describes
metal dynamics by taking as a starting point a past year when metal dynamics can be assumed to
be "pristine", that is uninfluenced by anthropogenic activities. Chemical processes influencing
metal dynamics, solid/solution partitioning and fixation into soil solid phases, are described in a
manner that seeks to reduce the number of variables required, while maintaining consistency with
mechanistic understanding of the underlying processes. Soil hydrology intended as annual volume
of drainage from a soil layer is specified, not modelled, and this allows for a range of complexity
in the specification of hydrology, for example considering annual variation in drainage or fixing
to a single value over time. Similarly, key properties influencing metal dynamics such as the pH
value of the soil solution, dissolved organic carbon (DOC) concentration, SOM content and soil
erosion rate, may be fixed to single values or varied annually.
The objective of this study was to apply the IDMM-ag model to a well-characterized location with
ideally no site-specific calibration in order to assess its performance at larger scale. The hypothesis
was that, if the model is successfully applied at field scale with no need of calibration, it might be



used at larger scale as well, provided adequate inputs. We therefore assessed the dynamics of Zn,
Cu, Pb and Cd in the topsoil of the ZOFE agricultural long-term trial in Switzerland, where the
organic amendments farmyard manure, sewage sludge and green waste compost have been
incorporated into soil for more than 60 years. Then, we applied the IDMM-ag model under the
different organic amendment managements in order to (*i*) test the capability of the IDMM-ag
model to reproduce the long-term changes in the labile pools of the TEs in the four treatments, and
(*ii*) predict the labile and soluble concentration trends of the TEs in the next 100 years and assess
any possible risk for the ecosystem and human health. At the current state of the art, large-scale
modelling could be informative for both broad trends in TEs concentration and information on the
time taken for a particular soil type to reach concentrations at which environmental risks occur.



**2.  Materials and Methods**
**2.1 The study site**
The Zurich Organic Fertilization Experiment (ZOFE) is an agricultural long-term plot trial started
in 1949 by the Swiss Federal Agricultural Research Institute (Agroscope) at Zurich-Reckenholz,
Switzerland, to compare different fertilization schemes in an 8-year crop rotation: 1) winter
wheat/intercrop, 2) maize, 3) potato, 4) winter wheat/intercrop, 5) maize, 6) summer barley, 7)
clover grass ley, 8) clover grass ley (Oberholzer et al., 2014). Ploughing has been carried out to a
depth of at least 20 cm, from north-to-south and *vice versa*, alternating the direction of adjacent
passes (Figure 1). The site is located at 420 m a.s.l., the annual precipitation has been 1054 mm in
the long-term average, and the mean annual temperature 9.4°C. The soil is a carbonate-free, loamy
(14% clay) Luvisol (IUSS, 2006), with a SOC content of 1.43% and a pH value of 6.5 at the
beginning of the experiment. The field trial consists of twelve treatments replicated in five blocks
in a systematic block design (Figure 1), and the same cultivation techniques and plant protection
have been applied to all the treatments. In the present study we investigated the following four
treatments: the control (NIL#1) with no fertilization and no amendment, the farmyard manure
(FYM#2) with application of 5 t ha$^{-1}$ of organic matter every second year, the sewage sludge
(SS#3) with application of 2.5 t ha$^{-1}$ of organic matter every year, and the green waste compost
(COM#4) with application of 2.5 t ha$^{-1}$ of organic matter every year.

**2.2 Trace element time series**
The NIL, FYM, SS and COM soils (top 20cm) were sampled from the Agroscope ZOFE soil
archive and analysed for total and EDTA-extractable concentrations of Zn, Cu, Pb and Cd. Soils
sampled were from years 1972, 1979, 1982, 1991, 1995, 2000, 2003, 2007 and 2011. Before 2011,





the samples from the five replicate plots per treatment had been bulked, so that the variability
between replicate plots could not be assessed. The archived samples comprised only the 2mm-
sieved fraction. To determine total soil TEs concentrations, sample extracts in 1 M nitric acid were
analysed by means of an inductively coupled plasma optical emission spectrometry (ICP-OES Dv
sequential Perkin Elmer Optima 2000). The EDTA-extractable pools were obtained with the
extraction protocol described by Quevauviller (1998) followed by ICP-OES analysis. For each
metal we took the ratio of the EDTA-extractable concentration and total concentration in the same
year to define the metal lability as a measure of the biogeochemically-available fraction at that
point in time.
Samples of farmyard manure of years 2011 and 2014, sewage sludge of years 2008 and 2012 for,
and compost of years 2011, 2013 and 2014 were also analysed for total and EDTA-extractable
concentrations of Zn, Cu, Pb and Cd with the extraction procedures described above.

### 2.3 The Intermediate Dynamic Model for Metals with lateral mixing

The IDMM-ag predicts annual concentrations of metals within topsoil and fluxes of metal from
soil due to porewater leaching and crop uptake, and distinguishes between a pool of labile
(geochemically active) TEs, comprising dissolved and adsorbed metal, and a non-labile (aged)
pool that accounts for chemically-less reactive and physically-protected solid forms of metals
(Figure 2A). The labile metal pools are partitioned into dissolved and adsorbed forms assuming
chemical equilibrium. A Freundlich-type isotherm (Groenenberg et al., 2010) is used to describe
the relationship between free TEs and adsorbed TEs ions, and the relationship between free TEs
ions and TEs complexes in the porewater is computed using WHAM/Model VI (Tipping, 1998).



Transformations between labile and aged pools are assumed to follow first-order kinetics. Labile
TEs may reversibly transfer into a 'weakly aged' pool, from which metal may subsequently
transfer irreversibly into a 'strongly aged' pool that can transfer directly into the labile pool. The
IDMM-ag is driven by annual TEs input rates: all the metal inputs were considered fully labile and
added into the model labile pool. The model simulations start from a past year in which all metal
inputs are assumed to be natural, and where the soil is in steady state, i.e. metal input and output
fluxes balance (Tipping, 1998). In this study erosion was neglected in consideration of the site
geomorphological characteristics that make it negligible. Since the soil samples were relevant to
the homogenised ploughing depth, the soil was modelled as a single well–mixed layer of 20 cm.
Based on the experimental work of Mcgrath & Cegarra (1992) and McGrath (1987) on the
influence of ploughing on lateral mixing of soil metals across plot boundaries, and based on initial
IDMM–ag simulations, the effect of lateral mixing was explored through a model modification
that allowed the plots to be simulated as sets of strips, with annual exchange of soil and TEs across
the strips (Figure 2B). In particular, lateral soil exchange was assumed to occur at the margins of
the strips across a width of 0.2 m, equal to the ploughing depth. A sensitivity analysis was carried
out on the number of strips per plot from five to ten, in order to understand its impact on the
simulations.

### 2.4 Estimation of metal input fluxes for the model

In the absence of anthropogenic sources, the metal inputs occurring naturally via geogenic
deposition (mainly volcanic eruptions) and soil mineral weathering are considered to be in balance
with the output fluxes. The natural inputs were assumed to be constant over time. In analogy to
the work by Rieder et al. (2014) for a Swiss forest, the geogenic deposition was estimated by



averaging the metal enrichment factors reported for Zn, Cu, Pb, Cd by Shotyk et al. (2002) in deep
layers (deposited before 1905) of a peat bog in the Jura Mountains, Switzerland. Mean Enrichment
Factors (EFs) were converted to actual atmospheric depositions by using moss concentration data
described later for anthropogenic atmospheric deposition. The estimated geogenic deposition rates
were: $1.1 \cdot 10^{-6}$ mol m$^{-2}$ yr$^{-1}$ for Zn, $1.15 \cdot 10^{-6}$ mol m$^{-2}$ yr$^{-1}$ for Cu, $2.52 \cdot 10^{-6}$ mol m$^{-2}$ yr$^{-1}$ for Pb,
$9.74 \cdot 10^{-8}$ mol m$^{-2}$ yr$^{-1}$ for Cd. The required magnitude of the mineral weathering flux was estimated
by fixing it to be constant over the simulation period, and adjusting it to make the modelled labile
TEs concentrations in 1972 equal to the observed values. The fitted additional fluxes were used in
all simulations, assuming them to be the same for all the plots.
Rates of anthropogenic atmospheric deposition of Zn, Cu, and Cd before c.a. 1990 were estimated
by means of the EFs reported by Shotyk et al. (2002). For Pb, the EFs before 1990 were taken
from Weiss et al. (1999), averaging peat bog sites with annual precipitation similar to ZOFE. From
1990 to 2014, atmospheric deposition data were estimated from the metal concentrations in mosses
measured in Northern Switzerland sites (BAFU 2018). These concentration values were converted
into atmospheric deposition rates by means of the transfer functions reported by Thoni et al. (1996)
for Switzerland (Figure S1 in Supporting Information).
Since farmyard manure P content has been measured in ZOFE throughout the experiment, Zn and
Cu inputs were calculated by multiplying the Zn:P and Cu:P ratios with the actual P loading from
the manure relying on the data by Menzi and Kessler (2009), who reported stable average
concentrations of Zn and Cu per unit of P in 1100 FYM samples. The Zn:P and Cu:P ratios were
averaged from the values measured in the farmyard manure samples applied in 2011 and 2014 as
reported in Table 1. To take into account the strong reduction of the Zn and Cu contents in FYM
observed from 1999s as a consequence of the decreased supply of TEs in the animal forage (de



Vrie et al., 2004; Groenenberg et al., 2006; Menzi and Kessler, 1999), Zn and Cu inputs with
farmyard manure application were reduced by factors of 0.7 and 0.52, respectively, starting from
1999 onwards. The Pb and Cd inputs with FYM application were also calculated from the P
content, using Pb:P and Cd:P ratio values of 0.495 and 0.027, respectively, taken from the work
of Menzi and Kessler (2009) ,because the total Pb and Cd concentrations in the farmyard manure
samples from 2011 and 2014 (Table 1) were below the ICP-OES detection limit (Figure S2 in
Supporting Information).
The TEs concentrations measured in the green waste compost samples applied in 2011, 2013 and
2014 (Table 1) were averaged and assumed to be constant throughout the experiment. For Cd,
which was below the instrument detection limit, a value of 0.13 mg kg$^{-1}$ was used, based on a
nation-wide investigation of compost quality in Switzerland (Kupper et al., 2014) (Figure S2 in
the Supporting Information).
For the sewage sludge tTEs inputs, we followed two approaches. In a first approach termed '*Swiss*
*Sludge Trend*', we averaged the metal concentrations in the SS samples from 2008 and 2012 (Table
1) and assumed that they were representative for the period 2000-2014. For the 1975-2000 period,
the exponential decrease in metal contents reported by Kulling et al. (2001) for sewage sludge in
Switzerland was applied to the measured values; before 1975, the metal concentrations were kept
constant and equal to the values calculated in 1975. In the second approach termed '*Idealized*
*Trend*' we kept the measured metal concentrations constant for the period 1975-2014 and fitted
the inputs for the period 1965-1975 to match the peaks measured in the EDTA-extracted
concentration trends. Before 1965, the soil metal concentrations were considered negligible. The
metal loading time trends were determined by multiplying the metal concentrations, as estimated





according to the two approaches described above, by the amount of sewage sludge applied (Figure
S3 in the Supporting Information).

**2.5 Metal outputs and other driving variables**

The two routes for metal output were leaching and crop uptake. Average water drainage was
calculated from rainfall measured at local stations and evapotranspiration estimated with a locally
calibrated Primault equation. Since no porewater dissolved organic carbon (DOC) concentration
data were available, a constant value of 7 mg C l$^{-1}$ was assumed, a reasonable value for agricultural
soils with low SOC content as reported by De Troyer et al. (2014). Furthermore, a preliminary
sensitivity analysis showed that varying the DOC concentration in the plausible range of 7-12 mg
C l$^{-1}$ had little effects on the results, with minor increase for the fitted additional input flux in case
of higher DOC concentrations. Crop metal removal was assumed to be a function of crop biomass
(Figure S4 in the Supporting Information), as crop metal concentrations were assumed not to vary.
Crop yields have been measured in ZOFE yearly based on the harvest from a sub-plot in each plot.
Shoot biomass was estimated by scaling linearly with crop yields. The Zn, Cu, Cd concentrations
in winter wheat grains and shoots were measured at harvest in 2014 and 2015 and the average
values were taken to represent the respective metal contents over the entire duration of the
simulated experiment in the grains and shoots of wheat and barley. The TEs concentrations for the
other crops were estimated by previous reports (de Vries et al., 2008; EFSA, 2009, 2010; SAEFL,
2003; SCAN, 2003a,b)..
The IDMM–ag model uses SOM and porewater pH as key variables controlling metal
solid/solution partitioning, aging and speciation, and both soil pH values and SOC content data
were available for soils from all ZOFE plots since 1949. The SOM contents were estimated
assuming that SOC is 50% by weight of the measured SOM. Values of pH from aqueous extracts



were converted into porewater pH values according to de Vries et al. (2008). The SOM content
and porewater pH values were provided to the model for all available years and linearly
interpolated when missing (Figure S4 in the Supporting Information).

**2.6 Analysis of the soil and organic amendment FTIR and XRD**
The NIL, FYM, SS and COM amended soil samples from 1972 and 2011 were analysed by Fourier
Transform Infrared Spectroscopy (FTIR) and by X-Ray Diffraction (XRD) to detect compositional
changes in the soil organic and inorganic components. The DRIFT spectra were obtained using a
rapid-scan Spectrum-GX (Perkin Elmer, Monza, Italy) Fourier transform infrared spectrometer
(FTIR) in the mid-infrared spectral range (4000 to 450 cm$^{-1}$). The spectrometer was equipped with
a Peltier-cooled deuterated triglycine sulphate (DTGS) detector and an extended range KBr beam
splitter. Soil samples of 50 mg were placed in a stainless steel sample cup, located in a Perkin
Elmer diffuse reflectance accessory and scanned for 60 s. A silicon carbide (SiC) reference disk
was used as the background sample (Perkin-Elmer). The most noticeable pecks were attributed
according to (D'Acqui, Santi, Vizza, & Certini, 2015; Niemeyer, Chen, & Bollag, 1992) as reported
in Supporting Information. Chemometric PCA analysis were carried out by the Unscrambler X®
Version X 10.4 (Camo Software) with spectra pre-processed with Extended Multiplicative Scatter
Correction (EMSC). The X-ray diffractometry (XRD) investigation was conducted on randomly-
oriented powders of bulk soil by a Philips PW3830 X-ray diffractometer with CoKα radiation,
0.02° step size, and 1s step time each point over a 2θ range of 5-75°. The SS soil sample from
2013 was also analysed.



**2.7 Evaluation of long-term effects of organic amendment applications and concentration limits**

We assessed the long-term effects of organic fertilization under the ZOFE trial environmental conditions. After validating IDMM-ag against the measured EDTA-extractable data, the model was run in predictive mode for 100 years, starting from pristine conditions until year 2114. The lateral mixing effect was removed as we aimed at simulate a real agricultural scenario with bigger plots than in ZOFE. In particular, the following boundary conditions were applied: *i*) stable TE input rates *via* anthropogenic deposition and organic amendment applications as recorded in 2014 (geogenic deposition and weathering rates were kept constant as previously described), and the *'Idealized Trend'* approach to estimate the metal inputs with the sewage sludge application, *ii*) stable SOM and soil pH as recorded in 2014, stable crop yields as recorded in the last rotation before 2014, and *iii*) stable temperature and rainfall, no climate change was taken into account.

Measured total TEs concentrations were compared with threshold values expressed as soil total concentrations reported in the Swiss Ordinance Relating to Impact on the Soil (OIS, 1998). The soluble concentrations from OIS (1998) extracted with 0.01 M $NaNO_3$ solution were used as indicators of potential soil ecotoxicity, and assessed with the projected pore water concentrations from the long-term IDMM-ag simulations. The projected labile concentrations were compared against the critical limits calculated for each metal according to the method proposed by Lofts et al. (2005) and based on the free ion approach: basically the ecosystem critical limits, expressed as labile metal pools, are determined by soil solution pH value, SOM and multispecies fraction affected, which was set to 0.1 in this study.

**2.8 Statistical analysis**





All statistical analyses were carried out in R (version 3.5.0). The Mann-Kendall test (package
"Kendall") was used to assess monotonic trends in the TE time series. Increasing trends (Kendall's
tau statistic > 0) and decreasing trends (Kendall's tau statistic < 0) were considered significant
when the two-sided $P$-value was less than 0.05.  The "dplyr" package was used to calculate the
root mean squared error ($RMSE$ with the "rmse" function) of the simulated labile concentrations
versus the measured data. The linear correlation between metal simulations and measurements was
assessed with the Pearson correlation coefficient ($r$ with the "cor" function).



## 3. Results and Discussion

### 3.1 Temporal trends of total and labile TEs

Despite the continuous application of organic amendments, the TEs total concentrations had no significant ($P < 0.05$) accumulation patterns over time in the topsoil according to the Mann-Kendall trend test, except for Pb concentration that increased significantly ($P < 0.05$) in the NIL treatment topsoil due to the atmospheric deposition (Figure 3). The total concentration trends were decreasing for Zn in the SS treatment and for Cu in all the treatments ($P < 0.05$), with Cu that displayed the strongest decrease from 60–102 mg kg$^{-1}$ in 1972 to 30–57 mg kg$^{-1}$ in 1995. The concentrations measured in 1972 were clearly elevated when compared to Ballabio et al. (2018), where an average total Cu concentration of ca.17 mg kg$^{-1}$ was reported from more than 21000 topsoils of EU countries. The observed higher Cu concentration in 1972 could be ascribed to the previous historic application of Cu-based fungicides, although we have no information on the duration and rates of fungicide application. The rate of total Cu loss from the studied soil was larger than that expected by leaching which is in the order of 0.2-0.3 mg kg$^{-1}$ of Cu per year amounting to 5-8 mg kg$^{-1}$ in 35 years (Vulkan et al., 2000). Mixing of the top 20 cm of soil with less contaminated deeper soil due to bioturbation could have caused a 'dilution' effect in the topsoil (Jarvis et al., 2010; MullerLemans & vanDorp, 1996).

In the SS treatment the total Zn, Cu and Pb concentrations exceeded the thresholds of the Swiss environmental legislation (OIS, 1998) set to 150, 40 and 50 mg kg$^{-1}$ for Zn, Cu and Pb, respectively. The Cu concentrations exceeded the thresholds of the Swiss environmental legislation in the soils of NIL, FYM and COM treatments during the 1970s then returned fully compliant to the legislation in the 1990s, whereas the total Cd concentrations never exceeded the guide value of 0.8 mg kg$^{-1}$. For comparison, the temporal trends of total P concentrations in the



topsoils were also analysed, and they showed less variability than the TE concentrations, a
significant accumulation over time in the FYM and SS treatments ($P < 0.05$), not in the NIL and
COM treatments (Figure 3).
The TEs lability in the ZOFE soils, here defined as the ratio between the EDTA-extractable
concentration and the total concentration is shown in Figure 4A. The Zn lability increased
significantly ($P < 0.05$ according to the Mann-Kendall test) only in the FYM treatment. This could
be due to the fact that the EDTA-extractable Zn, Cu and Cd showed significant increasing trends
over time in the NIL and FYM treatments except for Cu in the FYM treatment (Figure 5S in
Supporting Information) as a consequence of labile metal inputs, but also to soil acidification and
SOM loss observed in the NIL and FYM treatments (Figure 4S in Supporting Information). The
lack of significant increases in EDTA-extractable pools for Zn, Cu and Cd in the COM soil could
be due to the increase of the pH value over time. The lability of Cu significantly increased ($P <$
$0.05$) in the NIL and FYM treatments due to the decrease of its total concentrations. In the SS soil
the EDTA-extractable TEs decreased significantly ($P < 0.05$) despite soil acidification and SOM
loss also occurred in this treatment (Figure 5S in Supporting Information). Whereas EDTA-
extractable trends paralleled similar trends of decreasing total concentrations in the SS treatment
for the other metals, this was not the cases for soil Pb. Therefore, Pb lability trend was decreasing.
The Cd lability was generally the highest of all the metals across the treatments which is consistent
with its known lower affinity for the soil inorganic and organic soil colloids (McBride, Richards,
Steenhuis, & Spiers, 1999).
The total TE concentrations of the organic amendments are reported in Table 1. The farmyard
manure and compost samples had comparable levels of total TE concentrations, with the green
waste compost presenting higher Pb concentrations (Table 1). The sewage sludge had higher total



TE concentrations than the compost and farmyard manure, explaining why the magnitude of all
total TE concentrations ranked in the order NIL< COM = FYM < SS in 2011, and SS amended
soil also showed P overfertilization. The analysed sewage sludge showed the highest variability in
TEs lability over time, with Zn lability varying from 0.39 to 0.15 and Cu from 0.48 to 0.22 in the
samples from 2008 and 2012, respectively (Figure 4B). The lower lability of Cu and Pb in the
organic amendments than in the topsoil could be ascribed to their stronger affinity for the organic
matter (McBride et al., 1999).

**3.2 Simulations of soil metal dynamics**
The IDMM-ag model was run to simulate the measured EDTA-extractable concentrations with the
model-derived labile metal pools. However, to force model agreement with the measured EDTA-
extractable metals for the NIL plots in 1972, we had to enter additional input rates of TE as constant
in time. The fitted additional inputs were: 2.5 mg m$^{-2}$ yr$^{-1}$ for Zn, 5.5. mg m$^{-2}$ yr$^{-1}$ for Cu, 0.35 mg
m$^{-2}$ yr$^{-1}$ for Pb and 0.07 mg m$^{-2}$ yr$^{-1}$ for Cd. This additional TE term, being a calibrated variable,
was attributed to mineral weathering, although the site history also suggests that there is likely to
be a contribution from inputs of fungicides to the labile Cu concentrations at the beginning of the
measurement period. Estimates of TE weathering rates of 0.001-0.86 mg m$^{-2}$ yr$^{-1}$ for Zn and 0.0-
0.039 mg m$^{-2}$ yr$^{-1}$ for Cd in Swiss agricultural soils have been reported (Imseng et al., 2018; Imseng
et al., 2019), which were consistent with the fitted values. However, there is a need for more
research on topsoil metal weathering rates and their contribution to determining labile metal
concentrations in order to reduce the overall uncertainty on historic TEs inputs.



Using five strips per plot for simulating the lateral mixing, and the two approaches '*Swiss Sludge*
*Trend*' and '*Idealized Trend*' for estimating the sewage sludge metal inputs, the IDMM-ag model
produced the labile concentration time trends presented in Figure 5. The simulations without lateral
mixing are shown only for reference (Figure 6S in Supporting Information), since their predictions
of the measured data were unsatisfactory. With the '*Swiss Sludge Trend*' approach, the measured
EDTA-extractable concentrations were well simulated for all TEs in the NIL, FYM and COM
treatments. Consideration of the lateral mixing improved the agreement with the measurements
for the FYM and COM treatments by predicting higher labile metal concentrations, as a result of
metal transfer from the adjacent SS plots. In fact, the introduction of the soil lateral mixing
levelled-off the concentration differences between adjacent plots through the redistribution of TEs
from the most enriched SS plots to the adjacent FYM and C plots. The improvement of the
simulations for the FYM and COM treatments supported the hypothesis that lateral mixing is a
major transfer process in these experimental soils. The simulations were not fully adequate for the
SS treatment, with r values of 0.22, 0.02, 0.02, 0.64 for Zn, Cu, Pb, Cd respectively, and a lack of
declining trends in EDTA-extractable soil metal concentrations observed over the measurement
period (Figure 5; Figure S6). However, for Cd the model predicted a slight decline in
concentrations from the 1990s onward, possibly be due to the fact that Cd showed the largest
differences in concentration among the treatments, making the influence of the lateral mixing more
pronounced.
The model outcomes for the SS treatment, where the initial measurements in 1972 were
underestimated by up to a factor of three, suggest that the historic TE inputs to this plot were
underestimated. To investigate this, we adjusted the time trends of the metal inputs to the SS plots
to match more closely the observed trends (the *'Idealized Trend'* approach). After adjusting the



inputs the model was able to reproduce the downward trends in the EDTA-extractable TEs
observed in the SS treatment, with r values of 0.75, 0.73, 0.92, 0.88 for Zn, Cu, Pb, Cd respectively
(Figure 5). However, the measured EDTA-extractable soil TE concentrations were slightly
overestimated in the FYM and COM treatments, particularly for Cu and Pb. In fact, the *'Idealized*
*Trend'* approach implied high metal inputs with the sewage sludge applications from the 1960s to
the 1980s (Figure S2). These TEs were then rapidly transferred from the SS plots to the adjacent
FYM and COM plots due to ploughing, leading to an overestimation of the labile TEs
concentrations. The NIL plots, which are not adjacent to the SS plots, remained unaffected by the
approach used to estimate the TE inputs with the sewage sludge. We therefore hypothesize that
the soil lateral mixing was not the only cause of the bump-shaped trends observed in the SS soil,
because this TE input approach led to an overestimation of the labile concentrations in the FYM
and COM soils.

**3.3 Lateral mixing sensitivity analysis**
Splitting each plot into strips and exchanging a fraction of soil between adjacent strips at each time
step gave satisfactory results with the following best-fit operational parameters (Figure 2): *i)* each
plot was split into five strips, ii) the margin of a strip that was swapped with the adjacent strip was
20cm wide. A sensitivity analysis was carried out varying the number of strips per plot from five
(each strip being 1m wide) to ten (each strip being 0.5m wide), but keeping the same margin of
20cm to be swapped. The '*Idealized Trend*' approach was used because it gave better data fitting.
Figure 6 shows the simulated labile concentrations of Zn, Cu, Pb and Cd in 2014 across a transect
comprising the plots NIL, FYM, SS and COM in series, to replicate the order in the repetition
blocks I, II and III (Figure 1). Increasing the number of strips per plot from five to ten resulted in





a more pronounced bell-shaped pattern, with less redistribution of the TEs from the SS to the
adjacent plots. In general, the higher the number of strips per plot the less the contribution of the
soil lateral mixing, so that an adjustment of the '*Idealized Trend*' approach would be required to
fit the decreasing trends of the metal EDTA-extractable concentrations in the SS treatment
consisting of a small metal enrichment before 1980 followed by a decline. The opposite effect was
observed when a lower number of strips per plot than five was used. Unfortunately, because no
detailed information of the TEs content time trend in the sewage sludge applied to ZOFE was
available, we could not optimize the number of strips per plot via data fitting. Furthermore, the
two parameters (the number of strips per plot and the margin of the plot to be swapped) are inter-
connected but we tested the variation of only the first of them, thus neglecting another potential
source of uncertainty which should be considered in a full sensitivity analysis.

**447    3.4 Soil spectroscopy analysis and long-term effects of organic amendment applications**

The first two components of the PCA of the FTIR  analysis of organic and inorganic components
of the  ZOFE soil samples from 1972 and 2011, and from 2013 only for the SS plots, covered 86%
of the total spectral variance, with PC1 and PC2 accounting for 52% and 34%, respectively. Soils
from 1972 and 2011 clustered separately, and the separation was mainly due to the organic C
content (Figure 7A). This was confirmed by the loading plot of FTIR peaks of organic matter (OM)
and quartz (Q) minerals (Figure 8B). For PC1 the OM peaks were positively and the Q peaks
negatively correlated, whereas the PC2 showed an opposite trend, with the Q peaks slight prevalent
than in PC1 (Figure 8B). The PC1 indicated that the soils underwent to SOM depletion over time,
and the PC2 signalled that all soils, except SS, had higher presence of sand (Q peaks) in 2011than
1972, possibly due to loss of fine materials like clays. Concerning the peculiar behaviour of the



SS soil along the PC2, the differential FTIR spectra of the soils from 1972, 2011 and 2013 (the
subtraction of the spectra of 2011 and 2013 from the spectra of 1972) resulted in a variable
composition of the SOM between the soil from 2011 and 2013, regardless of the short time interval.
The differential spectra of the SS samples from 1972-2011 evidenced a particular peak at 1040
$cm^{-1}$ associated to the functional group of polysaccharide-like compounds that was not present in
the differential spectra of the SS samples from 1972-2013. These compounds have potentially high
affinity for TEs (Geddie & Sutherland, 1993; Veglio et al., 1997).
The XRD spectra did not reveal differences among soil samples regardless of sampling years and
treatment (Figure S7, Supporting Information). This result indicate that the long term organic
amendments, including the sewage sludge, did not introduce exogenous minerals such as clay
minerals and Fe-(oxy)hydroxides that could have modified the TEs availability. Further work is
needed to determine whether there was any time trend in the organic matter composition of the
applied sewage sludge, as different sewage sludge quality can impact the lability of the freshly
incoming TEs, overcoming the simplifying hypothesis that TEs present in the organic amendments
are always fully labile. Such an improvement could explain the decreasing lability trend of Pb in
the SS soil and reduce the over-prediction of Cu and Pb in the FYM and COM treatments when
the '*Idealized Trend*'' approach was used with the lateral mixing. The importance of quantifying
the adsorption capability of the sewage sludge applied to agricultural soils for TEs dynamic
modelling, particularly for TEs with high affinity for the SOM such as Pb and Cu, was suggested
also by Bergkvist & Jarvis (2004).

**3.5 Long-term effects of organic amendment applications**

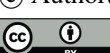



The long-term TEs labile concentrations (primary axis) and soluble concentrations in the pore
water (secondary axis) predictions are shown in Figure 9. As no soil lateral mixing was considered,
the labile TEs were predicted to accumulate in the SS plots beyond the measured concentrations;
on the contrary, the predicted concentrations in the neighbouring FYM and COM treatments were
predicted to become slightly lower than the measured values. The critical limits reported in Figure
9 (primary axis) for labile TE concentrations represent conservative estimations of biological
chronic toxicity in the soil. The critical limits were close to the predicted labile concentrations of
Zn and Cu in the NIL treatment though trends were declining on the long-term, and in the FYM
and COM treatments where the limits could be exceeded on the long-term due to continuous TEs
accumulation in the labile pool. In the SS treatment, Zn and Cu trends markedly exceeded the
critical limits, although Zn labile concentration were declining on the long-term, but Cu labile
concentration appeared to be stable over time. The Pb and Cd labile concentration trends were well
below the critical limits in all the treatments, with the exception of Cd in the SS treatment.
However, Cd was predicted to decline on the long-term in all the treatments, with the exception of
the COM treatment. The Cd decline due to its high mobility is in line with the long-term predictions
by Six & Smolders (2014) for the European agricultural soils. Globally, application of sewage
sludge at current rate of 2.5 t ha$^{-1}$ of organic matter every year appears unsustainable on the short
and long-term for Zn and Cu accumulation in the labile pool, with particular reference to Cu whose
labile concentration is projected to keep constantly high. Application of farmyard manure and
compost at current rates might result in chronic toxicity effects on the long-term.
The predicted pore water concentrations paralleled those of the labile concentration trends but with
a time-lag, and approximately by an order of magnitude below the Swiss legislation guide values
of 0.5 mg l$^{-1}$ for Zn, 0.7 mg l$^{-1}$ for Cu, 0.02 mg l$^{-1}$ for Cd, while no guide values are currently



provided for Pb, and noteworthy Zn in the SS treatment was the same order of magnitude of the
guide value. No calibration was done on the predicted soluble TE pools with measured data, so it
is not possible to conclude that the guide values set in the Swiss policy would be actually matched
in the future under stable conditions and current rates of soil organic amendment.



**4 Conclusions**


The IDMM-ag model provided adequate descriptions of the measured EDTA-extractable
concentration trends for the ZOFE long term field trial, when soil lateral mixing (site-specific) and
historic metal inputs adjusted to match observations were included. The labile concentrations of
Zn and Cu exceeded the critical limits, with potential toxic effects to 10% of the ecosystem species
in the SS amended soil, and it was predicted that their labile concentrations might exceed the
critical limits also in the FYM and COM amended soils on the long-term. Simulation of the EDTA-
extractable concentrations in the sewage sludge-amended plots and redistribution of the TEs to the
adjacent plots was possibly affected by the high variability of the sewage sludge organic matter
composition. This suggests that the metal input lability might vary and could be specified in the
TEs dynamic models to improve the simulations. The need to adjust the inputs with an additional
metal flux demonstrated that even for an experimental site with a well-known history, there may
be gaps in knowledge affecting all the models, regardless of their structure. At larger scale than
field-scale, such gaps are likely to exist also because the historic estimates might be inaccurate and
not greatly amenable to fine scale resolution.
Great improvements in TEs modelling will be brought by access to other datasets for testing the
model, quantifying the influence of bioturbation on the vertical redistribution of metals, assessing
metal weathering rates and their control factors, and analysing metal lability in the organic
amendments, particularly sewage sludge. Globally, the presented application of IDMM-ag is
promising for TEs dynamic simulations at field and larger scale, particularly if the current
limitation in the quality of the input data are overcome.





**Acknowledgements**
We would like to thank Lucie Gunst from Agroscope for making possible the resampling of soils
and amendments from the archive. Also, we gratefully acknowledge Dr. Susanna Pucci and Dr.
Luisa Andrenelli who carried out the TEs concentration analysis, Mr. Alessandro Dodero for
sample preparation for spectroscopic analysis and Dr. Alessandra Bonetti for FTIR analysis. This
research has been partially funded by the Doctoral Program of the DISPAA of the University of
Florence.





| | | Total Concentration [mg kg⁻¹] | | | | EDTA-Extracted Concentration [mg kg⁻¹] | | | |
|---|---|---|---|---|---|---|---|---|---|
| | | **Farmyard Manure** | | | | | | | |
| **Years** | **Zn** | **Cu** | **Pb** | **Cd** | **P** | **Zn** | **Cu** | **Pb** | **Cd** |
| **2011** | 109.5 | 23.59 | bdl* | bdl* | 6915 | 32.49 | 5.60 | bdl* | bdl* |
| **2014** | 158.9 | 27.78 | bdl* | bdl* | 8146 | 53.14 | 6.73 | bdl* | bdl* |
| | | **Sewage Sludge** | | | | | | | |
| **2008** | 447.5 | 165.80 | 24.76 | 3.18 | 16110 | 173.06 | 78.78 | 9.06 | 0.26 |
| **2012** | 715.0 | 301.20 | 35.40 | 7.00 | 28870 | 105.72 | 66.18 | 8.60 | 0.16 |
| | | **Compost** | | | | | | | |
| **2011** | 130.2 | 39.95 | 37.50 | bdl* | 2238 | 30.76 | 6.09 | 12.81 | bdl* |
| **2013** | 122.9 | 39.68 | 31.89 | bdl* | 2118 | 30.19 | 6.48 | 12.98 | bdl* |
| **2014** | 124.8 | 43.66 | 65.43 | bdl* | 2242 | 28.68 | 5.63 | 12.51 | bdl* |

*bdl=below detection limit (0.001 mg l⁻¹)

**Table 1:** Total and EDTA-extractable Zn, Cu, Pb, Cd and P concentrations relative to the total dry

matter of the organic amendment samples available.





**Figure Captions**

**Figure 1:** Experimental design of ZOFE with 12 treatments replicated in five blocks. Only the 4 treatments highlighted in grey were investigated: control (NIL #1), farmyard manure (FYM #2), sewage sludge (SS #3), and green waste compost (COM #4).

**Figure 2:** IDMM-ag model with lateral mixing. (A) Description of the model structure; (B) description of the soil lateral mixing implemented.

**Figure 3:** Total concentration time trends of Zn, Cu, Pb, Cd and P in ZOFE topsoils (20 cm) for the treatments NIL, FYM, SS and COM.

**Figure 4:** Lability time trends of Zn, Cu, Pb and Cd, expressed as the ratio between the EDTA-extractable concentration and total concentration, (A) in ZOFE topsoils (0-20 cm) for the treatments NIL, FYM, SS and COM; (B) in the organic amendment samples available.

**Figure 5:** Measured EDTA-extractable concentrations (•), simulated labile pool concentrations with the '*Swiss Sludge Trend*' approach (—), simulated labile pool concentrations with the *'Idealized Trend'* approach (--), of Zn, Cu, Pb, Cd in ZOFE topsoils (0-20 cm) for the treatments NIL, FYM, SS and COM. All the simulations are carried out with the lateral mixing effect. The two approaches differ in the way the sewage sludge metal content over time is estimated. RMSE = Root Mean Square Error; r = Pearson correlation.

**Figure 6:** Simulated labile concentrations of Zn, Cu, Pb and Cd in 2014 across a transect comprising the plots NIL, FYM, SS and COM in series with 5 and 10 strips per plot with the *'Idealized Trend'* approach.

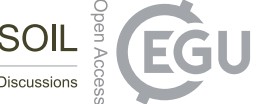

**Figure 7:** PCA analysis of the FTIR spectra from the NIL, FYM, SS and COM topsoil samples in
1972 and 2011. (A) PCA score plot with blue marks for samples from 1972 and red marks for
samples from 2011; (B) PCA loading plot with the most noticeable peaks classified as clay (Cl),
organic matter (OM) and quartz (Q).
**Figure 8:** Differential FTIR spectra of the SS samples: SS 2011 spectra subtracted from the 1972
spectra (red line), SS 2013 spectra subtracted from the 1972 spectra (blue line).
**Figure 9:** Measured EDTA-extractable concentrations (•), critical limits for labile concentrations
(primary axis), projections of labile concentrations (primary axis) and soluble concentrations
(secondary axis), for Zn, Cu, Pb, Cd from pristine conditions to 2114 in a real field experiencing
stable conditions and organic amendment applications as in ZOFE. All the simulations are carried
out without lateral mixing effect and with the *'Idealized Trend'* approach.








**Figure 1**







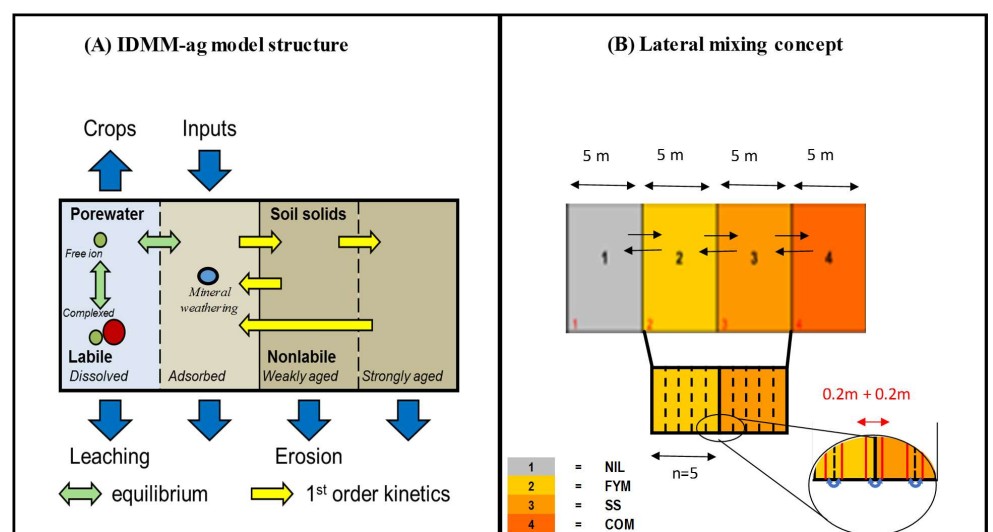



**Figure 2**





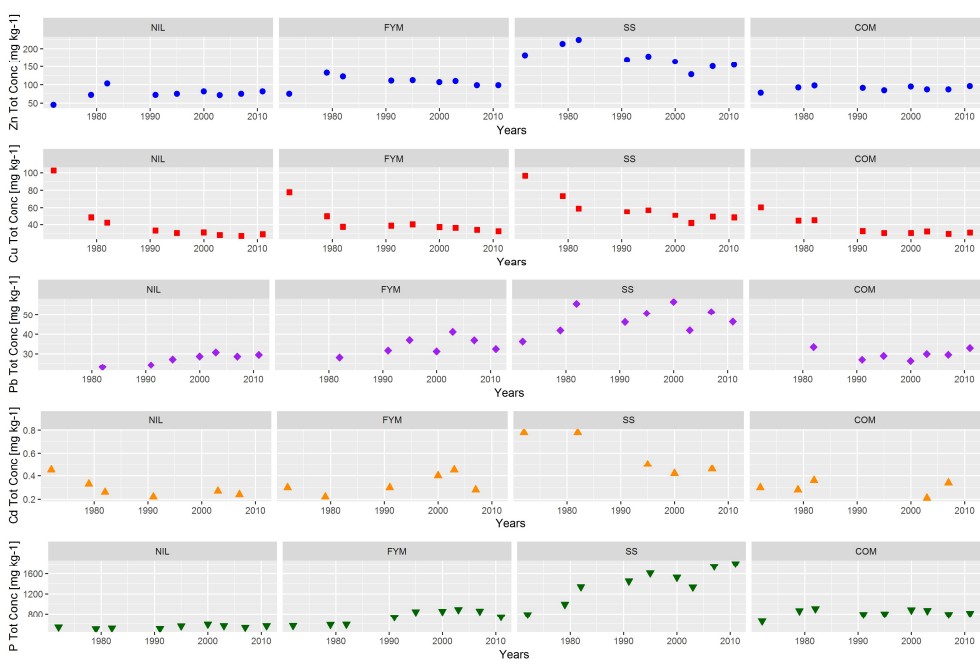

**Figure 3**



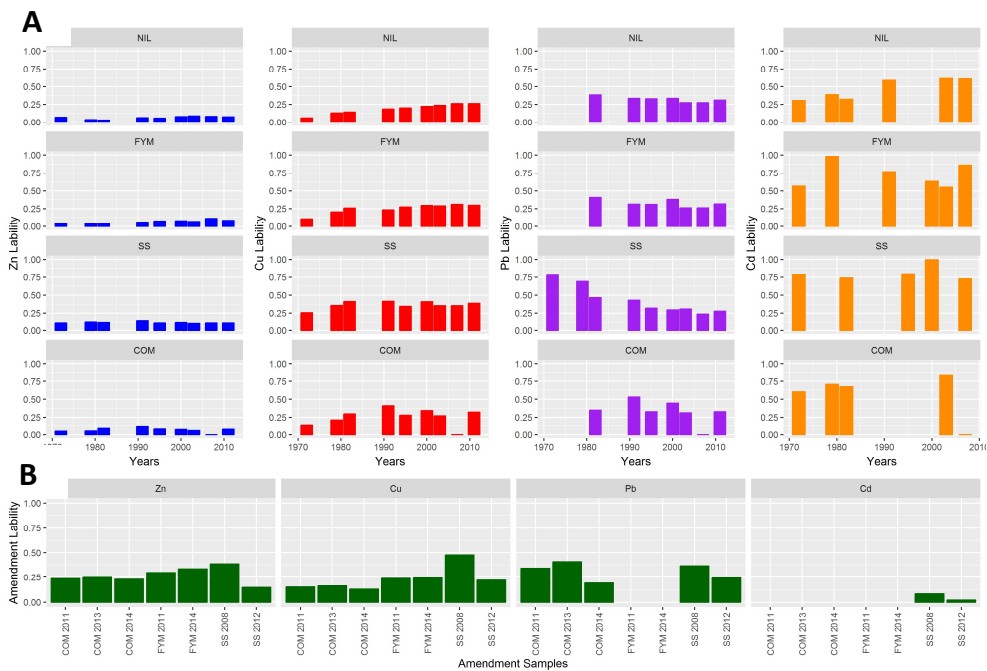

**Figure 4**



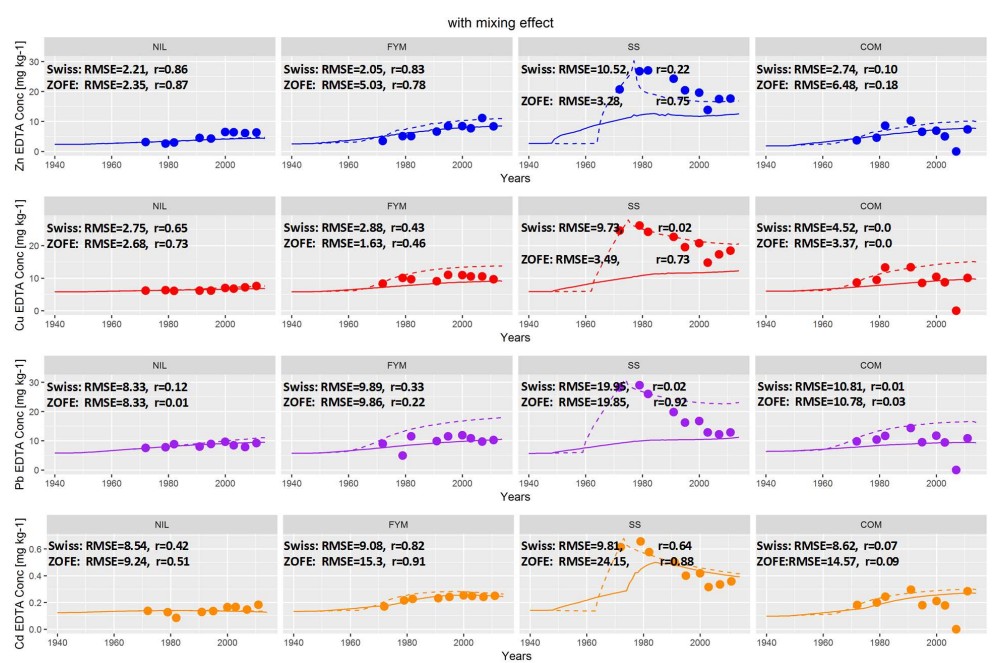


**Figure 5**





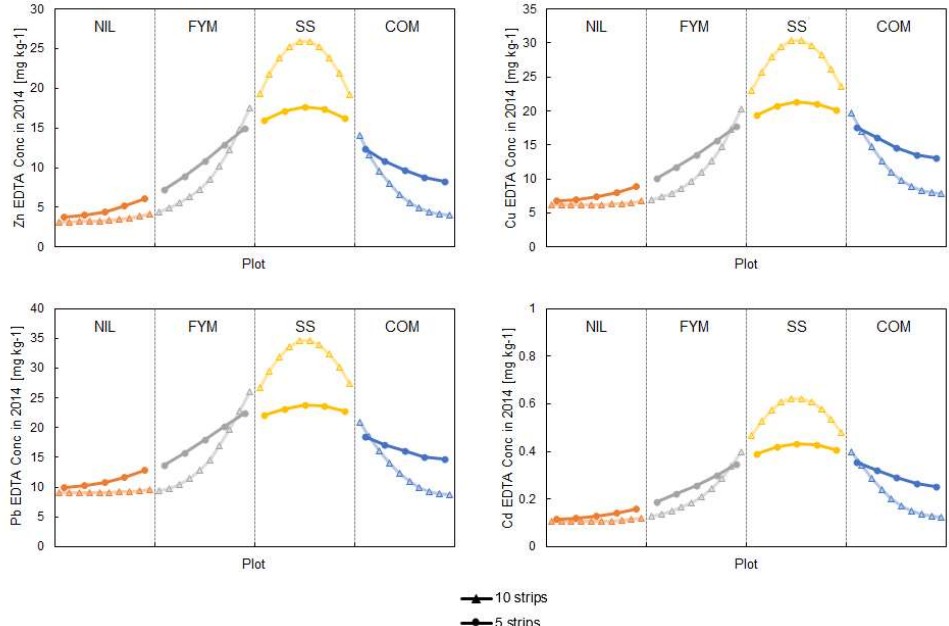


**Figure 6**













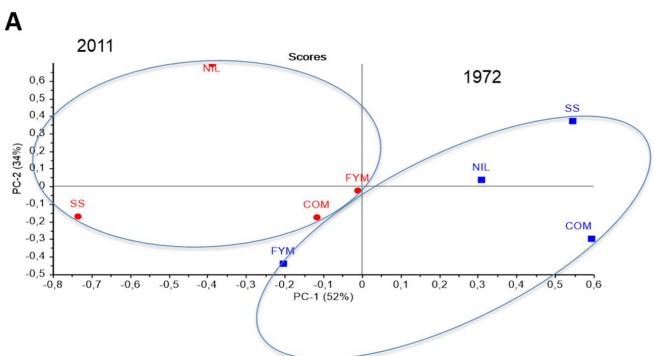

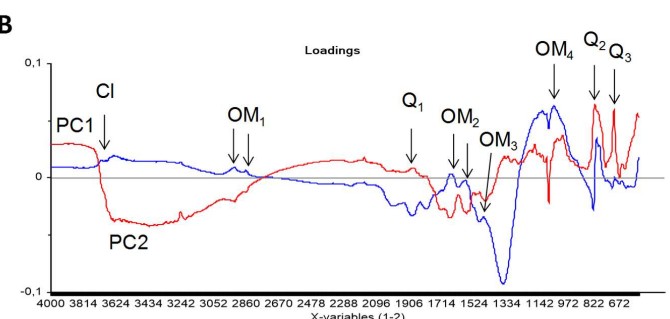


**Figure 7**















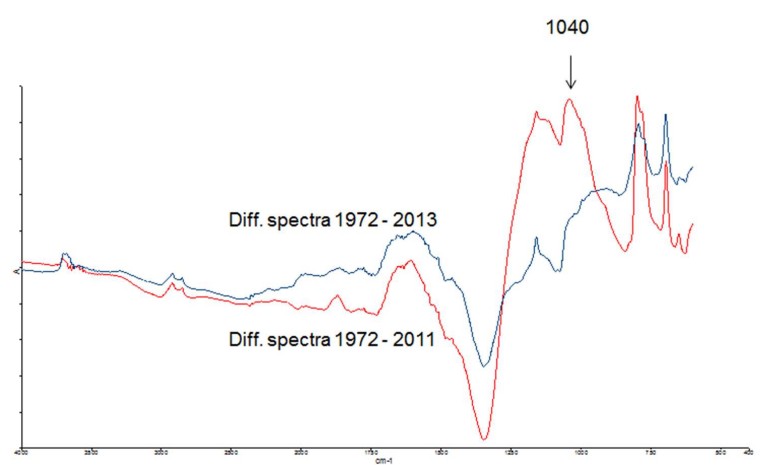


**Figure 8**








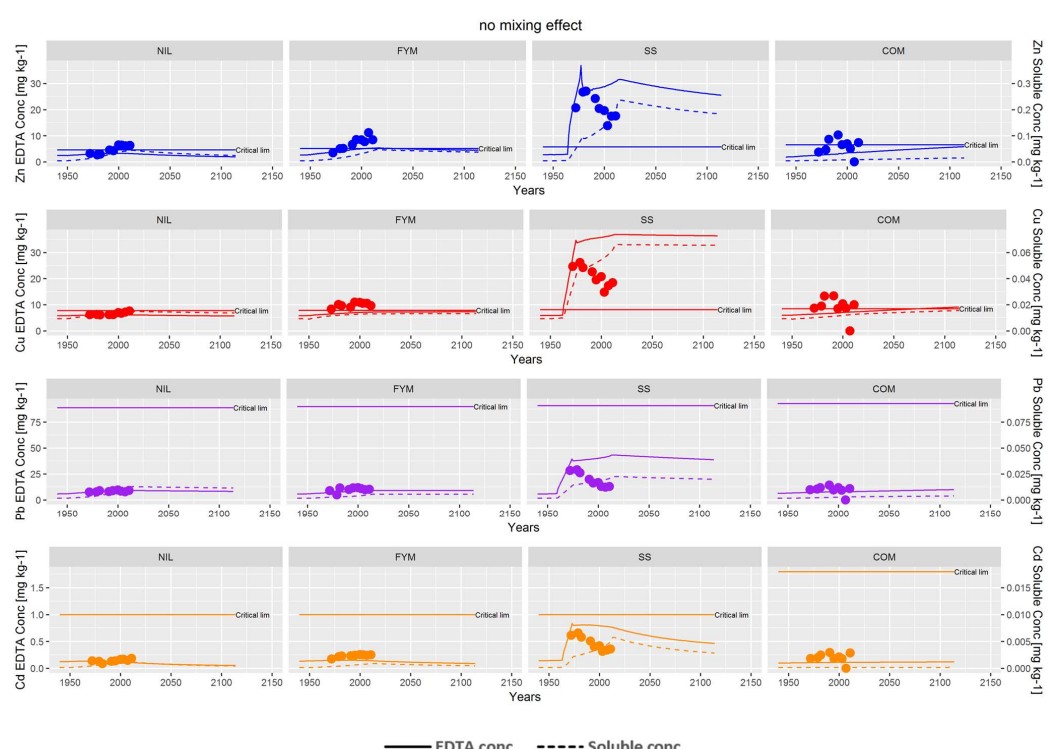



**Figure 9**



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
