# Peer review of "Modelling of long term Zn, Cu, Cd, Pb dynamics from soils fertilized with organic"

_SOIL, 2020_

## Referee Comment (RC1) · Anonymous Referee #1 · 21 Jun 2020

**Revision report soil-2020-21**

Modelling of long term Zn, Cu, Cd, Pb dynamics from soils fertilized with organic amendments.

The manuscript presented here deals with the model prediction/description of measured data that reflect the Zn, Cu, Cd and Pb dynamics in a long-term field trial amended with different organic amendments. Then, after evaluation, the used model is extrapolated to the future, to evaluate the possible risks of TEs by long-term application of organic amendments on agricultural fields. The advantage of this model is, according to the authors, that it has a restricted amount of input parameters. However, there are several issues with the manuscript in its current state.

1) The model description is not sufficient and scattered over different sections, which makes it difficult to link the different parts in the model. In addition, it should be much more clear which input parameters are needed for each part of the model (i.e., SOM and pH are likely used to calculate Kf's? What are input parameters for the WHAM + which complexes are considered?) + references for ageing parameters are not presented. At least an overview of considered reactions should be presented in the SI, to allow the reader to evaluate the restrictions of the model.

2) The measurements of the TEs lack quality control, the limit of quantification of each element and the relation with the measured data should be presented and, maybe most importantly, the Cd concentrations in the extracts are measured with ICP-OES, what could be troublesome regarding the known interferences during ICP-OES measurements with As in soil extracts.

3) The implementation of DOC in the model. Because no measurements were available, the DOC is fixed at 7 mg/L for all treatments (as in the beginning of the experiment) and remained constant, despite the application of different organic amendments. I am critical to this approach, due to the important effect of DOC to metal leaching. I would expect increasing (or changing between treatments) DOC concentrations over the years of the different organic amendment applications or at least increased DOC fluxes right after organic amendments that could increase metal leaching.

4) The artefacts associated with the experimental design of the ZOFE, namely plots touching each other + mixing of plots edges by ploughing. In the model evaluation, it appeared to be critical to introduce lateral mixing. This does not allow to evaluate the model without mixing, which is later used to extrapolate into the future. The lateral spread of TE concentration should be validated in a transect across plots in the ZOFE experiment by some new measurements, to underpin this model approach.

5) In the results, the treatment effects should be evaluated relative to the control data, to evaluate and contribute observed trends to the organic amendments solely, which is the scope of this study.

In addition, both the abstract and introduction lack quantitative data, the English writing could be improved and the final discussion of the results becomes difficult to follow starting from lines 408 to the end.

Further point-by-point comments are presented below.

**Abstract**

36: abbreviation of model

38: soil plots, different. Are there more organic amendments than the ones summed up here (particularly is not the best link word here)

39-40: link with previous sentence is missing, maybe this sentence can be declined here + don't start sentence with an abbreviation.

41: better provide quantitative measure of model performance.

41: abbreviation ZOFE

Wouldn't it be interesting to add the range of EDTA-extractable concentrations here in the abstract?

46: labile = EDTA-extractable? + provide projections, i.e. after XX years, concentrations could increase to YY.

**Introduction**

57-58: this sentence is too vague and is not really necessary here, can be skipped.

59-64: it would be interesting if you would add the concentration ranges at which the essential and the non-essential TEs become of a concern.

68: an

68-71: please rephrase to make the message more clear.

71-75: the link with the previous section is not clear. In addition, what do you mean with "limited natural availability of nutrient elements such as P"? Preferably start your sentence with the main message, for example "Organic amendments are considered to be more sustainable then inorganic mineral fertilizers, due to XX and YY".

77: isn't it just the transformation of the organic amendment to SOM that contributes to carbon sequestration as such, not additional C sequestration from atmosphere? If not, please explain more, but only if relevant for this study!

Actually, line 71-77 could be skipped from this introduction and you could go right at the possible introduction of TEs into soil by organic amendments, to keep the introduction to the point and relevant for this study.

77-81: please provide concentration ranges for the TEs in the different organic amendments.

83: fate? + please rephrase second part of sentence

86: you mean mobility (for solubility)?

87-89: please provide examples or more explanation of importance of TE speciation vs toxicity

93: direct reactions?

90: I think you can even say that it is not only useful, but even necessary

97-98: you already stated this in line 90-91

99-100: Please rephrase: I would not say that mechanistic models are site specific. Indeed, models that predict TE mobility and transfers over time by using as much as possible underlying physical and chemical mechanisms are likely only useful on a limited size scale, due to the high input needed, but they can be used at every site (when data are available), so they are not site specific. And what about empirical models?

102: what do you mean with behaviour?

102-106:  Ah, here you talk about empirical models. Please merge with text above and try to be more concise.

112-115: this is vague, could you specify more what variables you are talking about, and which mechanistic level of understanding is wanted.

Has this model been used already (and at what scale)? If yes, please provide the current state-of-the art of the performance on this model. What is the knowledge gap here for this study now?

121: IDMM-**ag**?

122: What is "larger scale" here?

123-124: "The hypothesis was that, if the model is successfully applied at field scale with no need of calibration, it might be used at larger scale as well, provided adequate inputs." -> can you test the second part of your hypothesis with this study? If not, please rephrase the hypothesis.

125: ZOFE?

131-133: please clarify sentence: "large scale", "broad trends", TE concentrations in soil? + rephrase final part of sentence.

**M&M**

150-152: for TE accumulation, the total applied amendment will likely be important, depending on the data collected on the TE content (per kg of material, per kg of OM,… ?).

160: 1M $HNO_3$ extractable metals are not total TE concentrations, please just write 1M $HNO_3$ extractable metals. And please also provide more experimental details, L:S ratio, extraction time. In addition, it would be interesting if this 1M $HNO_3$ extracted TEs could be related somehow to "total" element concentrations, measured by aqua regia or XRF or other, more standardized extraction protocols for total soil metal concentration.

162-163: please shortly describe, conc. of EDTA, L/S ratio, extraction time.

164: try to avoid the use of "we"

164-166: on what is the use of EDTA:1M$HNO_3$ extractable metals as a measure of lability based? Please clarify. Is this already used/tested? If so, please provide references.

What about quality control of these measurements, what is the limit of quantification of these methods and how do the measured soil samples relate to this? This could be described already here, or in the results section. In addition, the determination of Cd with ICP-OES in extracts from soil samples is troublesome due to the interferences with As, even at relative low As soil concentrations. For example see: "A comparison of reliability of soil Cd determination by standard spectrometric methods, M. McBride, JEQ 2011 (40, 1863-1865, doi: 10.2134/jeq2011.0096) and likely many other publications. Did you take this into account? If not, the reliability of the Cd measurements from this study can be severely questioned.

178: preferably write : free and adsorbed TE ions, in contrast to "free TEs"

176-179: please provide an overview of the Freundlich isotherms and TE complexes considered during this study in the supporting information. What are the input parameters for the Freundlich model (i.e. which extraction did you choose to represent the adsorbed fraction, and how does this relate to the adsorbed fraction represented in the initial models of Groenenberg (I think they used 0.43M $HNO_3$ acid extractable metals as "reactive" soil metals). In addition, what other soil properties were measured to calculate the $K_F$ by the transfer functions of Groenenberg and how are these soil

properties measured? What are the input parameters of the WHAM model and how are they measured?

180: please provide the first-order rate constants used for each element and explain wherefrom they are derived (references).

184: please provide the start year used in the calculations

186-187: please rephrase this sentence, it is not clear what is stated here

204-206: please clarify, not clear.

207-208: please be consistent in choice of unit

208-211: please provide the data of these fitted mineral weathering fluxes and compare with literature data, if possible.

217: on what are these transfer functions based? Please shortly describe.

220: P loading from the manure? You mean addition of P to the fields by manure application? You could quickly give the data here.

222: not clear, did you take the X:P ratio's from literature (=1100 data points) or from own measurements (= 2 data points)? Was P also measured in the FYM? That is not been described previously. In addition, how does the measured X:P ratio related to the literature reported?

227: derivation of these factors? One time decrease or decrease linearly with time? Not clear.

231: the detection limit in the caption of table 1 is expressed in mg/L while the concentration data are expressed in mg/kg. Please provide detection limit in mg/kg, to provide a clear idea of the lowest measureable concentration in the FYM. See comment above on the analytics of soils and organic amendments.

232: written like this, Figure S2 is about detection limits, which is not. Please rephrase.

238: **t**TEs + avoid "we"

245: which peaks? In the soils?

246: soil metal concentrations negligible?

Figure S3: in the swiss sludge trend, the Cd fluxes trend is deviating from the other metals. Why?

253-255: Ok at the start of the experiment, but I would expect increasing or changing DOC concentrations over the years of organic amendment applications (or between treatments) or at least DOC fluxes right after organic amendments. I think this approach (constant & low DOC over time) is not a good approach to simulate metal leaching over time in organic amendment treated plots (I assume this constant DOC is then the WHAM input?).

256: on what is this plausible range based?

257: minor increase for fitted additional input flux? Not clear. + what is minor?

261: not clear

261: measurements of plant material has not been presented + data for Pb? In addition, changes of plant TE concentrations with changes in labile TE concentration in soils were not considered?

267-268: It is not clearly explained how SOM and pH affect soild/solution partitioning, aging and speciation, because the input data/model description for the Freundlich, WHAM model and aging model are not well specified.

284: pecks?

290-291: why?

305-306: Please state how the soil total concentrations are/should be measured in this Swiss Ordinance and comment on own measurement data.

311-313: not clear + why 0.1?

**Results and discussion**

324: the trends in the organic amendment plots should be investigated relative to the trends in the control plots, to exclude all other enrichment/losses other than use of organic amendments, which is the core of this study. Then, the statistical analysis should be repeated on these relative data.

334-336: compare the measured Cu loss with the literature values + on what is this "expected" Cu leaching based.

348-364: same comment, compare treatment effects relative to control.

350-357: you should test correlations between the data to underpin these suggestions.

370: P-overfertilization? Based on what?

378-388: this was already (partly) described in the M&M section (and provides answer to above comments), please move this section to the M&M.

390: in the figures (also in SI), the ZOFE trend is mentioned. Is this the "Idealized trend"?

399-401: I have severe doubts of the applicability of the modelling results of TE dynamics to a realistic scale, as the experimental conditions of the field experiment are so specific, i.e., high TE concentration plots "contaminate" low TE concentration plots, so all the treatment effects are obscured by an experimental artefact (i.e. the plowing, the plots being so close to each other…). To be more clear -> the model was not capable to predict the measured concentrations, because the measured concentrations are affected due to the specific design and maintenance of the experimental plots, but such experimental plot are not relevant for real agricultural fields (i.e. narrow soil strips with different amendments that are influenced by lateral mixing), which makes the "fixing" of the model with the lateral mixing not really important for real situations. In addition, due to the fixing of the model by lateral mixing, the true performance of the model cannot be evaluated, and the extrapolation (done in figure 9) to real fields is questionable. However, I understand that this is related to the specific nature of this experimental design and that it is nevertheless worth to investigate the data available, due to the valuable information present from these long-term experiments. However, to verify the overall modelling approach (including lateral mixing and excluding it again to extrapolate the model), I think a simulated transect from figure 6 should be validated by measurements-> i.e. sample along a transect in the ZOFE experiment, measure labile concentrations and remodel for the sampling year.

402: I guess only the r-value of the Cd is significant? Provide significance of r-values.

408 and further: not clear anymore. Initial measurements underestimated? I've understand that these were fitted?

471-472: but the EDTA-extractable concentrations were measured? Couldn't this provide information of the "lability" of the TE input by the organic amendments…

---

## Author Comment (AC1) · 6 Jul 2020

We tackle below the five main observations raised by the Reviewer. 1) We recognize that the description of IDMM-ag given in the M&M is not sufficient to capture the model details, unless the cited references are checked. This was done partly by purpose to simplify the reading of an already-long manuscript. Therefore, we think that the suggestion given by the Reviewer to present a model overview, like in the form of a table, is very valuable and we will implement it. Also, we will edit the current layout of the model description in the M&M to give more details. 2) The Reviewer is making a good comment. In fact, ICP-OES could have interferences with As, particularly when this is present at medium/high concentration while Cd is close to the background value (see McBride. 20011. A comparison of reliability of soil Cd determi-

nation by standard spectrometric methods. J Environ Qual. 2011; 40(6): 1863–1869. doi:10.2134/jeq2011.0096). We didn't measure As concentration in the test field, but we made a comparison of the total metal concentrations obtained by extraction with aqua regia and analysis of the extracts with ICP-OES (by the lab that conducted all the measurements presented in the manuscript) and ICP-MS (in another lab in Zurich). In fact, ICP-MS is recognized to have greater sensitivity in Cd determination. The measurements of Cd concentrations in four different treatments obtained by ICP-OES and ICP-MS are in good agreement (see below). This test, which was carried out also for other trace elements, can be taken as a validation of the ICP-OES measurements, ruling out any possible interference with As. Please note that for determination of metal total concentrations in soils, the extractions have always been done with aqua regia; HNO3 was used only for extraction of the organic amendments. As to quality control of the ICP-EOS instrument, it is done on the calibration curve every 10 readings by measuring the TES concentrations in the blanks and in the standard sample at intermediate concentration of 1 ppm. The limit of quantification for every element is taken as 3 standard deviation of the readings from the blank samples. We will include more details of the analytical procedures in the M&M. 3) As correctly pointed by the Reviewer, we didn't have any DOC measurements available. The hypothesis that DOC concentration increases with organic amendment application is very likely, though the magnitude of this increase is dependent on many factors, such as application rate (maybe a threshold effect?), type of organic amendment, soil properties and local environmental factors. Since the inconsistency/scatter of the data in the literature (see a recent meta-analysis by Li et al (2019). "Effect of land management practices on the concentration of dissolved organic matter in soil: A meta-analysis". Geoderma 344 (2019) 74–81), it is very difficult to make an educated guess. However, since the reviewer's comments are valid, we have made additional analysis on the DOC contribution to TEs simulations. As a forward, the Reviewer suggests that DOC might peak after amendment application to slowly decrease until next application; this cycle is likely to happen, however IDMM-ag has an annual time step that requires an "average" DOC value to be specified. As a first test, we increased the DOC from 7 mgC/l to 33 mgC/l from pristine conditions to 2014 and applied this fixed concentration to all the treatments. These values come from the paper by De Troyer et al (2014). Factors Controlling the Dissolved Organic Matter Concentration in Pore Waters of Agricultural Soils. Vadose Zone J., which reports a large DOC dataset from European agricultural soils. In the ms we decided to apply a very low DOC value correspondent to the control treatment, characterized by big SOC loss and mechanical ploughing. Here we tested 33 mg C/l, the median value reported by De Troyer et al. (2014). The simulations with a fixed 33 mg C/l as DOC concentration did not differ from the ones with 7 mg C/l, with the only difference that we had to increase the additional input term (mineral weathering as we suggested in the ms) to fit the metal concentrations in 1972 for the control treatment. Except for Zn, the new fitted inputs are one order of magnitude bigger than the old ones: 3.99 mg m-2 yr-1 for Zn, 27.3 mg m-2 yr-1 for Cu, 3.52 mg m-2 yr-1 for Pb and 0.157 mg m-2 yr-1 for Cd. The higher values of this additional input term suggest that a five-fold increase in DOC concentration in the control treatment might be unrealistically high. Therefore, as a second trial we kept 7 mg C/l in the control treatment and in the other treatments before 1949 when the trial started. In particular, we applied a linear increase up to the following DOC concentration values in 2014: 56 mg C/l in the farmyard manure treatment, 70 mg C/l in the sewage sludge treatment, and 35 mg C/l in the compost treatment. We present below the results only with lateral mixing and the "idealized trend" as a simulation of the sewage sludge metal inputs. The model predictions slightly improved when increasing the DOC concentration with organic amendment applications: while the control treatment was barely affected, in FYM and COM the overestimation in labile concentrations, particularly for Cu and Pb, got smaller. Even in the sewage sludge treatment the TEs decreasing trends were better simulated. We can conclude that increasing the DOC concentration under organic amendment applications does not substantially modify the simulations, thus confirming the assumptions we made in the ms, but slightly improved them. So, we think it is valuable to add at least the second trial in the ms, presenting it as a test on DOC

concentrations since no DOC data are available to support the hypothesis. 4) Long-term experiments comparing different treatments are valuable sources of information, but here we showed a potentially unrecognized drawback: plots can be affected by soil mixing with the adjacent ones. This effect is hardly detectable unless elements present in trace concentrations are taken into account. Therefore, the lateral mixing was introduced to fit the data, since the 4 treatments considered were perfectly adjacent in 3 out of the 5 repetitions (in the other 2 repetitions the only compost treatment was separated from the "block"). In this sense, the lateral mixing is well represented by the considered treatments and collected data. Running another sampling campaign across the whole transect (12 treatments) is not feasible at the moment and, moreover, since some data are missing, it is not clear to know in advance the benefits of such an activity. However, we want to reinforce that lateral mixing serves the purpose of fitting the long-term trial data. The model, which is validated as long the simulations of historical data are satisfactory, should be run without lateral mixing when real fields are simulated. That's why we did not include the lateral mixing when projecting into the future the TEs accumulation as if ZOFE were a real field. 5) This comment is not very clear. In the pictures we showed the control treatment and the organic amendment treatments one next to the others because i) the control treatment takes part to the lateral mixing and it's worth showing the relevant simulations ii) absolute concentrations of TEs are useful on their own. However, we could show the concentrations of the organic amended treatments relative to the control treatment for the future projections. Regarding the other comments, we will review the English writing, include some quantitative data in the Abstract and Introduction and make clearer the last part of the paper. We thank the Reviewer for the useful suggestions throughout the paper.
* * *
**NIL**

**FYM**

**COM**

**NPK**

**Fig. 1.**

with mixing effect

Fig. 2.

---

## Referee Comment (RC2) · Anonymous Referee #2 · 7 Jul 2020

Manuscript title: Modelling of long term Zn, Cu, Cd, Pb dynamics from soils fertilized with organic amendments

General comments

This MS addressed the issue of the modelling of the contamination of agricultural soils by trace elements added by the long term application of organic residues and the toxicological and ecotoxicological consequences of such a contamination. This issue is relevant and of a timely interest. In particular, most of previous study were focused on the modelling of the change in total trace element concentration in soil, but the present MS is rather dedicated to the modelling of trace element availability (also called lability in the MS) in soil. This is clearly original and of an environmental interest to make the link with the potential toxicological and ecotoxicological consequences.

I have however a range of comments that to my point of view should be addressed before addressing this MS for publication in SOIL.

Major comments My first major comment is related to the general approach followed and to what is to my point of view the main conclusion of the paper. The author concluded (lines 509-511) that "the IDDM-ag model provided an adequate description of the measured EDTA-extractable concentration trends...". Looking at the figure 5, this is not so obvious. There are indeed several situations for which there is a clear discrepancy between experimental data and modelling (e.g. Zn-SS, Pb-SS, Cd-COM) and also other situations where the Swiss (e.g. Zn-FYM, Pb-FYM) or the ZOFE (e.g. Cu-SS) was alternatively the hypothesis which enables the model to have the best fit. In addition, these fits are based on the modelling approach considering lateral mixing. If the principle of lateral mixing is explained and if the uncertainty related to such a computation is discussed in the MS, there is not any validation of such a computation based on experimental data. In addition to that, there are a lot of uncertainty on some major flux of trace elements for a significant part of the field experiment history, particularly on trace elements added by organic residues. To consider whether the fits obtained were adequate or not, uncertainty in model parameters and input data should be considered and compared to the uncertainty in experimental data. Moreover, considering the question about the adequacy of the fits of EDTA-extractable concentration, it should be to my point of view necessary to show as the first step the fits obtained for total trace element concentration trends in soil. The idea is that if the total trace element trends are not adequately simulated, how the EDTA-extractable trend could be? The simulation of total trace element concentration trends seem to me even more necessary as the accumulation trend expected is not visible for most trace element and organic fertilisation modalities. In particular, total Cu concentration shows a strong decreasing trend that was attributed to the past application of Cu fungicide, then followed by a sharp removal of Cu from the top-soil layer. No convincing explanation is given for this as the authors said that they do not have information about Cu-fungicide applications and assumed that soil ploughing and bioturbation explained the dilution of Cu

concentration in soil without any simulation to support these strong assumptions. Without any other explanation, it seems that the Cu dataset is strongly biased and should, to my point of view, be removed from the MS.

My second major comment is related to the second major conclusion of the paper suggesting that Cu and Zn contamination in soil can be harmful to soil organisms. To my understanding, this conclusion is based on the methodology described lines 309-313. It is however really unclear how the related computation of critical limits was effectively achieved. I looked at the cited paper of Lofts et al. 2005, from which I supposed that the free ion approach was based on EDTA-extractable concentration, pH and SOM data. If I am right, I notably wonder how the natural background concentration of trace elements in the soil was considered as regard to the fact that this specific issue is addressed by Lofts et al. (2005). Also, this methodology was tested on two dataset from UK and North America. It is thus not obvious that the methodology is relevant for the specific case and consequently the specific application of the IDDM-ag model studied here.

Additional comments Lines 67-68 and 77-82: Sewage sludge is introduced differently from other organic residues (FYM and COM), notably because of the higher trace element concentration found in SS compared to FYM and COM. However, this is because the FYM and COM had relatively low trace element concentrations. For instance, I suppose that FYM is a cow manure. If a pig or poultry manure had been chosen, the concentration of several trace elements (Cu and Zn more particularly) would have been much higher and likely comparable to the concentrations observed in SS.

Line 199: To what refer the metal input? To the pool of total or available trace element?

Lines 199-211: It is not clear what is considered behind "geogenic input". It is also very surprising to use data on the weathering rate of deppe layers of peat bogs to estimate the weathering rate in the present field experiment where the soil is clearly not a peat bog.

Doc concentration was fixed at 7 mg C/L. The authors further argued that Doc variation between 7 and 12 mgC/L does not impact the leaching of TE. However, considering the large variation in SOM and pH in the different fertilization modalities, I am surprised that a larger range of Doc concentration was not expected. Several authors (e.g. Araujo et al. 2019, https://doi.org/10.1016/j.envpol.2018.12.070; Cambier et al. 2014, https://doi.org/10.1016/j.scitotenv.2014.06.105; Laurent et al. (2020, https://doi.org/10.1016/j.scitotenv.2019.135927) showed drastic change in Doc concentration in soil amended with organic residues. A way to estimate the initial Doc concentration and its likely evolution over time could have been to use the empirical multi-linear regression suggested by Romkens et al. 2004 (Derivation of Partition Relationships to Calculate Cd, Cu, Ni, Pb, Zn Solubility and Activity in Soil Solutions; Alterra: Wageningen, 2004; p 75).

Line 303: The choice to fix pH and SOM in soil at the value found in 2014 for predictive modelling is really disputable, when considering how these two parameters are strongly impacted by the long-term applications of organic residues and particularly in the context the field experiment studied as showed in figure S4. This point should at least be discussed.

Lines 324-338 and 348-364: These two paragraphs are really too descriptive and speculative. As related to my first main comment, the simulation of total trace element concentration trends seem a prerequisite to assess the adequacy of the model used. But, as far trace element availability in soil is concerned, some (usually found in the literature, but nevertheless strong) hypotheses on the soil parameters explaining the change over time in the EDTA-extractable concentration of trace elements. These hypotheses should be checked, for instance by looking for multi-linear regression between trace element EDTA-extractable concentration or lability and some important parameters such as the input of trace metal in soils, total trace metal concentration in soil, SOM and pH in soil.

Lines 425-445: Basically, the consideration of lateral mixing is interesting. However,

the comparison of simulations with and without lateral mixing should be showed clearly (at least in supporting information) to support the conclusion that accounting for lateral mixing is important.

Section 3.4: It is really unclear to me what is the added value of the FTIR and XRD datasets. To my point of view, these datasets should be removed.

---

## Author Comment (AC2) · 4 Aug 2020

**Referee #1**

Below we answer to the main points raised by the Referee.

1) We recognize that the description of IDMM-ag given in the M&M is not sufficient to capture the model details, unless the cited references are checked. This was done partly by purpose to simplify the reading of an already-long manuscript. Therefore, we think that the suggestion given by the Referee to present a model overview is very valuable. In particular, we will review the model description in the M&M and add the following paragraph in the SI.

[Figure]

The IDMM simulates a topsoil as a single, fully–mixed soil layer and computes concentrations of metals associated with the soil and in the soil porewater on an annual basis. The soil layer has a defined depth and comprises fine earth material (the soil material itself), coarse matter (stones) and pore space, which is partly filled by water. Annual gains and losses of metal (mol m$^{-2}$ yr$^{-1}$) are computed by a flux balance:

$$\Delta M = M_t + F_{input} + F_{weath} - F_{drain} - F_{crop} \tag{S1}$$

where $\Delta M$ is the change in metal pool (mol m$^{-2}$), $M_t$ is the metal pool in the soil before calculation (mol m$^{-2}$) and $F_{input}, F_{weath}, F_{drain}, F_{crop}$ are respectively the fluxes of metal into the soil in external input and weathering of metal the coarse fraction, and losses in porewater drainage and due to cropping.

Within the soil, metal is subdivided into a number of forms according to Figure 2A. These are:

1. Metal in porewater, comprising the free metal ion and metal complexed to solution ligands, including dissolved organic matter (DOM);

2. Adsorbed metal, comprising metal reversibly adsorbed to binding sites on the surface of the fine earth material. The sum of the adsorbed metal and the metal in porewater is the 'labile'

or 'geochemically active' metal, considered measurable by extraction using a strong ligand or dilute acid (e.g. 0.1M EDTA, 0.43M $HNO_3$).

3. Two pools of 'aged' metal. This is metal in the fine earth that is considered 'fixed' within the solid phase and only slowly exchangeable with the labile pool. The two pools are characterised by relatively fast and slow exchange kinetics and are termed the 'aged' and 'mineral' pools.

Chemical speciation of the labile metal, including its distribution between the solid phase and porewater, is handled by equilibrium, while exchanges of metal among the labile, weakly aged and strongly aged pools are handled by kinetics.

*Speciation and distribution of the labile metal pool*

The equilibrium speciation and solid–porewater distribution of the labile metal pool is computed annually by a combination of empirical modelling and application of the WHAM/Model VI speciation code (Tipping, 1998). Empirical modelling, following the approach of Groenenberg et al. (2010), is used to derive the relationship between the free metal ion concentration in the porewater and the adsorbed metal concentration, as a function of key soil properties:

$$log\, K_{f.m} = log\{M_{ads}\} - n \cdot log[M^{2+}] = \gamma_1 + \gamma_2 \cdot pH_{pw} + \gamma_3 \cdot log\{SOM\}. \qquad (S2)$$

Here $log\, K_{f.m}$ is a Freundlich–type partition coefficient, defined as $log\{M_{ads}\} - n \cdot log[M^{2+}]$ where $\{M_{ads}\}$ is the adsorbed metal concentration (mol g$^{-1}$), $[M^{2+}]$ is the free metal ion concentration in porewater (mol dm$^{-3}$) and $n$ is a fitted constant in the range zero to unity. The terms $pH_{pw}$ and $\{SOM\}$ are the porewater pH and the soil organic matter content (% w/w) respectively, and the $\gamma$ terms are fitted constants. The model is optimised by fitting to the error in $log\, K_{f.m}$; this drives a set of constants that provide consistent computation of $\{M_{ads}\}$ from $[M^{2+}]$ and vice versa. The constants used are taken from Groenenberg et al. (2010) and are shown in **Table S1**.

**Table S1.** Parameters for adsorbed metal–free ion relationships, adapted from Groenenberg et al., 2010

| Metal | $\gamma_1$ | $\gamma_1$ | $\gamma_1$ | n |
|---|---|---|---|---|
| Cu | -6.37 | 0.64 | 0.87 | 0.57 |
| Zn | -4.67 | 0.46 | 0.84 | 0.84 |
| Cd | -5.71 | 0.41 | 0.91 | 0.70 |
| Pb | -6.46 | 0.96 | 1.35 | 0.84 |

The relationship between the free metal ion and dissolved metal in the porewater is handled by WHAM/Model VI. Inputs to the model comprise the porewater pH and dissolved concentrations of Na, Mg, Ca, Cl and $SO_4$, and the DOM concentration. These variables are all specified. An initial adjustment is made, whereby either the concentration of Ca, or those of Cl and $SO_4$, are adjusted to achieve charge balance at the specified pH. The speciation of Al is handled using the approach of Tipping (2005) to estimate the activity of $Al^{3+}$ in the porewater on the basis of the pH and DOM

concentration. The speciation of FeIII is handled by assuming the porewater $Fe^{3+}$ to be in equilibrium with $Fe(OH)_3$, having a standard solubility constant of 2.5 and a standard enthalpy of -102 kJ mol$^{-1}$.

The equilibrium constants used are shown in Table S2 and Table S3.

Metal binding to DOM is simulated by assuming it to comprise 65% fulvic acid. Each metal has two binding constants, (i) log $K_{MA}$ which is used to derive binding constants for carboxylic and phenolic binding sites, and (ii) $\Delta LK_2$, which is used to generate constants for high affinity binding sites.

**Table S2**. Solution equilibrium parameters for Ca, Al, FeIII and carbonate.

| Equilibrium | log $K^\ominus$ | $\Delta H^\ominus$ (kJ mol$^{-1}$) | Reference |
|---|---|---|---|
| $H^+ + CO_3^{2-} \rightleftharpoons HCO_3^-$ | 10.329 | -14.899 | Nordstrom et al. (1990) |
| $2H^+ + CO_3^{2-} \rightleftharpoons H_2CO_3$ | 16.681 | -24.008 | Nordstrom et al. (1990) |
| $Mg^{2+} + H^+ + CO_3^- \rightleftharpoons MgHCO_3^+$ | 11.4 | -11.59 | Nordstrom et al. (1990) |
| $Mg^{2+} + CO_3^{2-} \rightleftharpoons MgCO_3$ | 2.98 | 11.34 | Nordstrom et al. (1990) |
| $Mg^{2+} + SO_4^{2-} \rightleftharpoons MgSO_4$ | 2.37 | 19.04 | Nordstrom et al. (1990) |
| $Al^{3+} + OH^- \rightleftharpoons AlOH^{2+}$ | 9.01 | -6.44 | May, Helmke, Jackson (1979) |
| $Al^{3+} + 2OH^- \rightleftharpoons Al(OH)_2^+$ | 17.87 | -15.40 | May, Helmke, Jackson (1979) |
| $Al^{3+} + 4OH^- \rightleftharpoons Al(OH)_4^-$ | 33.84 | -45.44 | May, Helmke, Jackson (1979) |
| $Al^{3+} + SO_4^{2-} \rightleftharpoons AlSO_4^+$ | 3.2 | 9.6 | Izatt, Eatough, Christensen, Bartholomew (1969) Sillen and Martell (1971) |
| $Al^{3+} + 2SO_4^{2-} \rightleftharpoons Al(SO_4)^-$ | 5.11 | 13.0 | Izatt, Eatough, Christensen, Bartholomew (1969) Sillen and Martell (1971) |
| $Ca^{2+} + H^+ + CO_3^- \rightleftharpoons CaHCO_3^+$ | 11.44 | -3.64 | Nordstrom et al. (1990) |
| $Ca^{2+} + CO_3^{2-} \rightleftharpoons CaCO_3$ | 3.22 | 14.85 | Nordstrom et al. (1990) |
| $Ca^{2+} + SO_4^{2-} \rightleftharpoons CaSO_4$ | 2.30 | 6.90 | Nordstrom et al. (1990) |
| $Fe^{3+} + OH^- \rightleftharpoons FeOH^{2+}$ | 11.81 | -12.39 | Nordstrom et al. (1990) |
| $Fe^{3+} + 2OH^- \rightleftharpoons Fe(OH)_2^+$ | 22.33 | -40.25 | Nordstrom et al. (1990) |
| $Fe^{3+} + 3OH^- \rightleftharpoons Fe(OH)_3$ | 29.44 | -63.97 | Nordstrom et al. (1990) |
| $Fe^{3+} + 4OH^- \rightleftharpoons Fe(OH)_4^-$ | 34.4 | -90.17 | Nordstrom et al. (1990) |
| $Fe^{3+} + SO_4^{2-} \rightleftharpoons FeSO_4^+$ | 4.04 | 16.36 | Nordstrom et al. (1990) |
| $Fe^{3+} + Cl^- \rightleftharpoons FeCl^{2+}$ | 1.48 | 23.43 | Nordstrom et al. (1990) |
| $Fe^{3+} + 2Cl^- \rightleftharpoons FeCl_2^+$ | 2.13 | – | Nordstrom et al. (1990) |

**Table S3.** Solution equilibrium parameters for Cu, Zn, Cd and Pb

| Equilibrium | log $K^\ominus$ | $\Delta H^\ominus$ (kJ mol$^{-1}$) | Reference |
|---|---|---|---|
| $Cu^{2+} + OH^- \rightleftharpoons CuOH^+$ | 6.48 | – | Sunda and Hanson (1979) |
| $Cu^{2+} + 2OH^- \rightleftharpoons Cu(OH)_2$ | 11.78 | – | Sunda and Hanson (1979) |
| $Cu^{2+} + H^+ + CO_3^- \rightleftharpoons CuHCO_3^+$ | 14.62 | – | Mattigod and Sposito (1979) |
| $Cu^{2+} + CO_3^{2-} \rightleftharpoons CuCO_3$ | 6.75 | – | Smith and Martell (1976) |
| $Cu^{2+} + 2CO_3^{2-} \rightleftharpoons Cu(CO_3)_2^{2-}$ | 9.92 | – | Smith and Martell (1976) |
| $Cu^{2+} + SO_4^{2-} \rightleftharpoons CuSO_4$ | 2.36 | 8.8 | Smith and Martell (1976) |
| $Cu^{2+} + Cl^- \rightleftharpoons CuCl^+$ | 0.4 | 6.7 | Smith and Martell (1976) |
| $Zn^{2+} + OH^- \rightleftharpoons ZnOH^+$ | 5.04 | – | Baes and Mesmer (1976) |
| $Zn^{2+} + 2OH^- \rightleftharpoons Zn(OH)_2$ | 11.1 | – | Baes and Mesmer (1976) |
| $Zn^{2+} + H^+ + CO_3^- \rightleftharpoons ZnHCO_3^+$ | 13.12 | – | Mattigod and Sposito (1979) |
| $Zn^{2+} + CO_3^{2-} \rightleftharpoons ZnCO_3$ | 4.76 | – | Mattigod and Sposito (1979) |
| $Zn^{2+} + SO_4^{2-} \rightleftharpoons ZnSO_4$ | 2.38 | 6.3 | Smith and Martell (1976) |
| $Zn^{2+} + Cl^- \rightleftharpoons ZnCl^+$ | 0.4 | 5.4 | Smith and Martell (1976) |
| $Cd^{2+} + OH^- \rightleftharpoons CdOH^+$ | 3.92 | – | Baes and Mesmer (1976) |
| $Cd^{2+} + 2OH^- \rightleftharpoons Cd(OH)_2$ | 7.65 | – | Baes and Mesmer (1976) |
| $Cd^{2+} + H^+ + CO_3^- \rightleftharpoons CdHCO_3^+$ | 11.83 | – | Parkhurst and Appelo (1999) |
| $Cd^{2+} + CO_3^{2-} \rightleftharpoons CdCO_3$ | 4.37 | – | NIST (2003) |
| $Cd^{2+} + 2CO_3^{2-} \rightleftharpoons Cd(CO_3)_2^{2-}$ | 7.26 | – | NIST (2003) |
| $Cd^{2+} + SO_4^{2-} \rightleftharpoons CdSO_4$ | 2.46 | 9.6 | Smith and Martell (1976) |
| $Cd^{2+} + Cl^- \rightleftharpoons CdCl^+$ | 1.98 | 1.3 | Smith and Martell (1976) |
| $Cd^{2+} + 2Cl^- \rightleftharpoons CdCl_2$ | 2.6 | 3.8 | Smith and Martell (1976) |
| $Pb^{2+} + OH^- \rightleftharpoons PbOH^+$ | 6.29 | – | Baes and Mesmer (1976) |
| $Pb^{2+} + 2OH^- \rightleftharpoons Pb(OH)_2$ | 10.88 | – | Baes and Mesmer (1976) |
| $Pb^{2+} + 3OH^- \rightleftharpoons Pb(OH)_3^-$ | 13.94 | – | Baes and Mesmer (1976) |
| $Pb^{2+} + CO_3^{2-} \rightleftharpoons PbCO_3$ | 7.2 | – | Buffle, Chalmers, Masson, Midgley (1988) |
| $Pb^{2+} + 2CO_3^{2-} \rightleftharpoons Pb(CO_3)_2^{2-}$ | 10.5 | – | Buffle, Chalmers, Masson, Midgley (1988) |
| $Pb^{2+} + SO_4^{2-} \rightleftharpoons PbSO_4$ | 2.75 | – | Smith and Martell (1976) |

| Equilibrium | log $K^\ominus$ | $\Delta H^\ominus$ (kJ mol$^{-1}$) | Reference |
|---|---|---|---|
| $Pb^{2+} + Cl^- \rightleftharpoons PbCl^+$ | 1.59 | 18.4 | Smith and Martell (1976) |
| $Pb^{2+} + 2Cl^- \rightleftharpoons PbCl_2$ | 1.8 | – | Smith and Martell (1976) |

**Table S4.** Constants for metal binding to fulvic acid in WHAM/Model VI. All from Tipping (1998) except where noted.

| Metal | log $K_{MA}$ | $\Delta LK_2$ |
|---|---|---|
| Mg | 1.1 | 0.12 |
| Al | 2.5 | 0.46 |
| Ca | 1.3 | 0 |
| FeIII[a] | 2.6 | 2.20 |
| Cu | 2.1 | 2.34 |
| Zn | 1.6 | 1.28 |
| Cd | 1.6 | 1.48 |
| Pb | 2.2 | 0.93 |

[a] log $K_{MA}$ from Tipping, Rey-Castro, Bryan, Hamilton-Taylor, 2002.

*Exchange between labile, aged and mineral pools*

Metal exchanges between the labile, aged and mineral pools are handled by a first order kinetic schema as introduced by Xu, Lofts and Lu (2016). The schema allows the following transfers of metal among the pools: (i) labile to aged, (ii) aged to labile, (iii) aged to mineral, and (iv) mineral to labile. These transfers are described by the kinetic constants $k_{f,a}$, $k_{f,b}$, $k_{f,m}$ and $k_{b,m}$ respectively. These are summarised in Table S5.

Table S5. Kinetic constants for metals transfers among the labile, aged and mineral pools.

| Metal | $k_{f,a}$ | $k_{f,b}$ | $k_{f,m}$ | $k_{b,m}$ | Reference |
|---|---|---|---|---|---|
| **Cu** | $10^{-2.5+10^{-3.3}e^{pH_{pw}}}$ | $10^{-2.1+10^{-3.5}e^{pH_{pw}}}$ | $10^{-5}$ | | [a] |
| **Zn** | $10^{-4.2+0.26pH_{pw}}$ | $10^{-3.2}$ | $10^{-5}$ | | [a] |
| **Cd** | $10^{-2.9+0.18pH_{pw}}$ | $10^{-1.6}$ | $10^{-5}$ | | [a] |
| **Pb** | $10^{-6.3+0.51pH_{pw}}$ | $10^{-2.6}$ | $10^{-5}$ | | this study |

[a] Xu, Lofts and Lu (2016)

Aging constants for Pb were derived by analysis of an ongoing long term experiment, comprising four UK soils. Soils were spiked with lead and incubated with maintenance of temperature and moisture content (55% of water holding capacity). Samples were periodically taken for quantification of lead spike lability using isotopic dilution. Fitting results are shown in Figure S1.

[Figure]

**Figure S1.** Aging of lead spikes in four UK soils and model fits using the expressions and parameters in Table S5.

**References**

Baes, C.F., Mesmer, A.E. (1976). *The Hydrolysis of Cations*. John Wiley & Sons, New York.

Buffle, J., Chalmers, R.A., Masson, M.R., Midgley, D. (1988). *Complexation reactions in aquatic systems : an analytical approach*. Ellis Horwood, Chichester.

Groenenberg, J. E., Romkens, P. F. A. M., Comans, R. N. J., Luster, J., Pampura, T., Shotbolt, L., de Vries, W. (2010). Transfer functions for solid-solution partitioning of cadmium, copper, nickel, lead and zinc in soils: derivation of relationships for free metal ion activities and validation with independent data. European Journal of Soil Science, 61(1), 58-73. doi:10.1111/j.1365-2389.2009.01201.x

Izatt R. M., Eatough J. J., Christensen J. J. and Bartholomew C. H. (1969) Calorimetrically determined log K, deltaH_0 and deltaS_0 values for the interaction of sulphate ion with several bi- and ter-valent metal ions. J. Chem. Soc. (A) 47-53.

Mattigod, S.V., Sposito, G. (1979). Chemical Modeling of Trace Metal Equilibria in contaminated soil solutions using the computer program, GEOCHEM. In: *Chemical Modeling in Aqueous Systems: Speciation, Sorption, Solubility*. Ed. E. Jenne. ACS Symposium Series 93, American Chemical Society, Washington DC.

May H. M., Helmke P. A., Jackson M. L. (1979) Gibbsite solubility and thermodynamic properties of hydroxy-aluminium ions in aqueous solution at 25°C. Geochimica et Cosmochimica Acta, 43, 861-868.

NIST (2003). *NIST Standard Reference Database 46 – NIST critically selected stability constants of metal complexes: version 7.0*. National Institute of Standards and Technology, US Department of Commerce.

Nordstrom, D. K., Plummer, L. N., Langmuir, D., Busenberg, E., May, H. M., Jones, B. F., Parkhurst, D. L. (1990) Revised chemical equilibrium data for major water-mineral reactions and their limitations In: *Chemical modeling of aqueous systems II*. Eds. D. C. Melchior, R. L.Bassett. ACS Symposium Series 416, p. 398-413, American Chemical Society, Washington DC.

Parkhurst, D.L., Appelo, C.A.J. (1999). *User's guide to PHREEQC (version 2) – a computer program for speciation, batch-reaction, one-dimensional transport, and inverse geochemical calculations*. US Geological Survey, Denver, Colorado.

Sillen L. G. and Martell A. E. (1971) Stability Constants of Metal-Ion Complexes, Suppl. No. 1. Chemical Society, London.

Smith, R.M. and Martell, A.E. (1976). *Critical Stability Constants, Vol. 4: Inorganic Complexes*. Plenum Press, New York.

Sunda, W.G., Hanson, P.J. (1979). Chemical Speciation of Copper in River Water: Effect of Total Copper, pH, Carbonate, and Dissolved Organic Matter. In: *Chemical Modeling in Aqueous Systems: Speciation, Sorption, Solubility*. Ed. E. Jenne. ACS Symposium Series 93, American Chemical Society, Washington DC.

Tipping, E. (1998). Humic Ion-Binding Model VI: An improved description of the interactions of protons and metal ions with humic substances. Aquatic Geochemistry, 4(1), 3-48. doi:Doi 10.1023/A:1009627214459

Tipping, E. (2005). Modelling Al competition for heavy metal binding by dissolved organic matter in soil and surface waters of acid and neutral pH. Geoderma, 127, 293–304. doi:10.1016/j.geoderma.2004.12.003

Tipping, E., Rey-Castro, C., Bryan, S.E., Hamilton-Taylor, J. (2002). Al(III) and Fe(III) binding by humic substances in freshwaters, and implications for trace metal speciation. Geochimica et Cosmochimica Acta, 66(18), 3211-3224. doi: 10.1016/S0016-7037(02)00930-4

Xu, L., Lofts, S., & Lu, Y. (2016). Terrestrial ecosystem health under long-term metal inputs: modeling and risk assessment. Ecosystem Health and Sustainability, 2(5), e01214. doi:10.1002/ehs2.1214

2) ICP-OES could give interference of As with Cd measurements, particularly when As is present at medium/high concentrations while Cd is close to the background values (see *McBride. 2011. A comparison of reliability of soil Cd determination by standard spectrometric methods. J Environ Qual. 2011; 40(6): 1863–1869. doi:10.2134/jeq2011.0096*). Unfortunately, we did not measure As concentrations in the field, though we do not expect high enrichment of As in ZOFE soil. However, as a validation of the analytic procedure, we compared the total metal concentration time trends obtained by extraction in aqua regia and analysis of the extracts with ICP-OES with the total metal concentrations obtained from another laboratory in Zurich which analysed the same plots in 2013

and partly in 2014 using extraction in aqua regia and analysis with ICP-MS. In fact, ICP-MS is recognized to have greater sensitivity in Cd determination. We show below the comparison for Cd: measurements from ICP-OES and ICP-MS are in good agreement. This test rules out any possible interference with As. Please note that for determination of total concentrations in the soils, the extractions were done with aqua regia and not with 1 M HNO3 as erroneously indicated in the paper, which was used only for extraction of the organic amendments.

[Figure]

[Figure]

As to the quality control of the ICP-EOS measurements, it is done on the calibration curve every 10 readings by measuring the TES concentrations in the blanks and in the standard sample at the intermediate concentration of 1 ppm. The limit of quantification for every element is 3 standard deviations of the readings from the blank samples. Therefore, we propose to modify the paragraph in the M&M as follows:

The NIL, FYM, SS and COM soils (top 20cm) were sampled from the Agroscope ZOFE soil archive and analysed for total and EDTA-extractable concentrations of Zn, Cu, Pb and Cd. Soils sampled were from years 1972, 1979, 1982, 1991, 1995, 2000, 2003, 2007 and 2011. Before 2011, the samples from the five replicate plots per treatment had been bulked, so that the variability between replicate plots could not be assessed. The archived samples comprised only the 2mm-sieved fraction. To determine total soil TEs concentrations, sample extracts in aqua regia were analysed by means of an inductively coupled plasma optical emission spectrometry (ICP-OES Dv sequential Perkin Elmer Optima 2000). The EDTA-extractable

pools were obtained with the extraction protocol described by Quevauviller (1998) followed by ICP-OES analysis. *Total metal concentration trends obtained with ICP-OES were compared with one-point-in-time measurements from the same plots carried out in other laboratory with ICP-MS, to rule out any interference of As with Cd readings (McBride, 2011). Quality control of the ICP-OES was done on the calibration curve every ten readings by measuring the TES concentrations in the blank samples and in the standard sample at the intermediate concentration of 1 ppm. The limit of quantification for each element was calculated as three standard deviations of the blank readings.* For each metal we took the ratio of the EDTA-extractable concentration and total concentration in the same year to define the metal lability as a measure of the biogeochemically-available fraction at that point in time.

Samples of farmyard manure of years 2011 and 2014, sewage sludge of years 2008 and 2012 and compost of years 2011, 2013 and 2014 were also analysed for total *(extraction with 1 M nitric acid)* and EDTA-extractable concentrations of Zn, Cu, Pb and Cd.

3) The Referee raises an important point on the role of the porewater DOC concentration, and we acknowledge that we have not justified the assumptions made regarding its concentrations. Therefore, we produce to introduce the following considerations into the ms:
Data on porewater DOC concentrations from arable soils are scarce and contradictory. To the best of our knowledge, there are no consistent data from long-term experiments. There are a small number of meta-analyses (for example Li et al., 2019; de Troyer et al., 2014), but they are not ideal, because: 1) they do not contain data on soils amended according to all the management approaches taken in ZOFE, so some residual assumptions on DOC would be required; 2) we could not find any data on long-term time trends of field DOC concentrations under arable soils; 3) DOC concentrations obtained from laboratory soil extractions differ from data collected directly from the field using lysimeters, with the latter usually showing lower concentrations; if this is the case, most of the available DOC data are likely to be overestimates of 'true' field concentrations and thus bias the model results. We compared the predictions of the equation suggested by Referee #2 to estimate DOC from pH and SOM (Derivation of Partition Relationships to Calculate Cd, Cu, Ni, Pb, Zn Solubility and Activity in Soil Solutions; Alterra: Wageningen, 2004; p 75) with data from two studies that measured DOC sampled in field using lysimeters:

[Figure]

There is a consistent trend to overestimation of the observed DOC concentration (up to at least two orders of magnitude) and no relationship between observed and predicted DOC concentrations. Therefore, we strongly conclude that the Alterra equation should at best be applied with great care. Application of the equation to the ZOFE plots produced predicted DOC concentrations in the range 60-80 mg/dm3, which we contend, on the basis of the chart above, is highly likely to overestimate DOC and thus not be useful for our purposes.

In conclusion, we think that sources of reliable DOC concentrations for agricultural fields are effectively missing, and so should be identified as key research priorities. Therefore, we suggest that (i) this knowledge gap should be openly confronted and emphasised in the ms; (ii) given this lack of knowledge, the most pragmatic approach is to carry out a systematic sensitivity analysis within a plausible range of DOC concentrations, incorporating also time trends. We will do a mini-review of field measurements of annual DOC fluxes for temperate grassland sites (before 1949) and arable with/without organic amendments (after 1949) and we will perform simulations with the minimum, maximum and midpoint of our established range of DOC fluxes. To incorporate the time trends in the sensitivity analysis, we propose to apply to each plot a simplified SOC model (for example, a two-pool model like the one in Menichetti et al. (2016), which was previously applied to ZOFE), assuming that DOC correlates with the decomposition fluxes out of these pools. Clearly, this estimation of DOC is affected by the assumptions of C input decomposability when shifting the site management from grassland to arable in 1949; in addition, this approach does not include the effect of soil acidification, which is reported to control the adsorption/desorption od DOC. However, it could be considered a first estimation of DOC time trend.

We have already made some preliminary checks and the main conclusions are the following: increasing the DOC concentration in the control treatment has the effect of increasing the additional flux of metals that we hypothesized to be mineral weathering from the coarse fraction – this is to expected as the assumed DOC and weathering fluxes are not independent since the

DOC controls the predicted soil labile concentration at steady state. Also, the impact of increasing five-fold the DOC concentration in the control treatment has a modest effect on the simulated labile concentrations (always <10%). On the other hand, increasing the DOC concentrations in the organic-amended treatments relative to the control treatment has the effect of lowering the metal concentration time-trends in these treatments, thus improving the overall simulation; yet, the effect is also modest.

Finally, the Referee suggests that DOC might peak after amendment application to slowly decrease until next application; this "cycle trend" is likely to happen, however since IDMM-ag has an annual time step, such short term effects are not modelled and an annual average DOC concentration, corresponding to the annual DOC flux, is what the model requires.

References:

Li et al. (2019). Effect of land management practices on the concentration of dissolved organic matter in soil: A meta-analysis". Geoderma 344 (2019) 74–81

De Troyer et al (2014). Factors Controlling the Dissolved Organic Matter Concentration in Pore Waters of Agricultural Soils. Vadose Zone J.

Menichetti et al. (2016.). Parametrization consequences of constraining soil organic matter models by total carbon and radiocarbon using long-term field data. Biogeosciences, 13, 3003-3019.

4) Long-term experiments comprising different treatments are valuable sources of information, but here we showed a potentially unrecognized drawback: plots can be affected by soil mixing with ploughing (plots have a separation space, but eventually it is too narrow for mechanical ploughing). This effect is hardly detectable unless elements present in trace concentrations are taken into account. Therefore, lateral mixing was introduced to check whether we could fit the data with realistic mixing coefficients. The 4 treatments considered were perfectly adjacent in 3 out of the 5 repetitions (in the other 2 repetitions the compost treatment only was separated from the "block"), so we can argue that the lateral mixing is well represented by the considered treatments and collected data. Clearly, the specific conditions in the long-term experiments cannot be extrapolated to real field conditions, and this is why we did not include the lateral mixing when projecting into the future the TE accumulation. Running another sampling campaign across the whole transect (12 treatments) is not feasible at the moment and, moreover, since some data are missing, it is not clear the real benefits we will get out of such an activity.

5) In the pictures we showed the control treatment and the organic amendment treatments one next to the others because i) the control treatment takes part to the lateral mixing and it's worth showing the relevant simulations ii) absolute concentrations of TEs are useful on their own. However, we could show the metal concentrations of the organic amended treatments relative to the control treatment for the projections into the future.

Regarding the other comments, we thank the Referee for the valuable comments and we will take them in the ms.

---

## Author Comment (AC3) · 4 Aug 2020

**Referee #2**

We thank the referee for acknowledging the interest in the ms.

- We acknowledge the fact that there is great uncertainty in the input data, as well as on the mechanisms governing TEs accumulation in the soil; decoupling and/or quantifying these uncertainties is out of the scope of the ms. Yet, the approach followed to quantify the inputs was able to reproduce the observed trends with lateral mixing and under the "Idealized Trend" assumption (also called ZOFE assumption in Figure 5, which we recognize could be misleading and should be corrected for homogeneity). We pointed out that the metal inputs with sewage sludge was perhaps not fully cached, and we did some investigation to understand the reason if it. Regarding the implementation of the lateral mixing, please see answer #4 to Referee 1. Said this, we recognize that simulation adequacy would require additional considerations, therefore we won't use this term in the ms.

- The IDMM is mainly designed to simulate labile metal concentrations, but we agree that it would be informative to show the total metal concentration simulations as well. Therefore, we will include them, but we propose not to disregard the Cu data. Indeed, we do recognize that the hypothesis of fungicide application and bioturbation/soil removal is not conclusive, but we should also consider that: i) the Cu labile concentration trends are plausible and a valuable source of information; ii) we should give evidence in the literature of those data which could not be explained satisfactorily, for future research; iii) there is no obvious reason why Cu total concentrations should be biased, while the total concentrations of the other metals should not be so. Therefore, we propose to show the total concentration simulations for Zn, Cu, Cd, Pb, leaving open the question of the large decrease of Cu, which the model will not simulate.

- We will use a slightly different but more standardized approach to estimate trace element critical limits. This time we will apply: *Lofts et al. (2004). Deriving Soil Critical Limits for Cu, Zn, Cd, and Pb: A Method Based on Free Ion Concentrations. Environ. Sci. Technol. 2004, 38, 3623-3631.* Furthermore, in the future projections we will show the variation of the critical limits over time together with pH and SOM changes. We agree with the Referee that the background concentrations should be subtracted for the calculation of the critical limits; therefore, we propose to subtract the trace element concentrations estimated by the model before 1949 (the start of the experiment).

- The organic amendments are applied at different frequencies and quantities in order to introduce the same (estimated) amount of organic matter, so that the comparison is done on the same basis. Since the TEs concentrations had not been assessed before this work, they did not contribute to the decision of applying different rates of organic amendment. Said this, the Referee is right that the FYM is from cow and the comment is valid that FYM from different sources could have much higher TEs concentrations. We will rephrase the sentence.

- By geogenic deposition we mean the natural concentration of TEs in the atmosphere, i.e. due to eruptions from volcanoes, which give rise to TEs deposition well before anthropogenic activities became prevalent. In fact, this deposition would be detectable even at "pristine conditions". This is why we used data from deep peat bog layers to estimate this natural deposition of TEs (please note that we did not use peat bog data to estimate mineral weathering!).

- Regarding DOC, please see the answer #3 to Referee 1.

- The Referee is right that even after >65 yr of soil management, none of the plots here considered have reached a new equilibrium condition: pH and SOC are still decreasing, sometimes at lower rates than initially. Therefore, keeping fixed pH and SOC can be a crude assumption. We propose to apply a SOC model to predict the future SOC (and DOC) changes over time (see answer #3 to

Referee 1). For pH, since its value will depend upon a number of factors, not least the speciation and cycling of added N, we propose to use simple extrapolation of the observed trends.

- Menichetti et al. (2016.). Parametrization consequences of constraining soil organic matter
- models by total carbon and radiocarbon using long-term field data. Biogeosciences, 13, 3003-3019.
-
- We will delete the description of the measured data and give space to the additional work that these comments have raised. Though it would be interesting to apply a multi-linear regression to explain the measured data, we feel it would be out of the scope of this ms, which is focused on model application and future predictions of TEs bioavailable concentrations.
- The spectroscopic analysis is useful for two important perspectives: i) confirming the general reduction of organic matter in soil for long term treatments in all samples; ii) confirming the importance to know the nature of organic material in terms of high affinity for TE's and its possible consequence in affecting the model.
  The X-Ray Diffraction analysis on soil samples give the possibility to establish that the organic treatments have not introduced exogenous mineral material, especially, in the case of the sewage sludge application. Differently, there are no other data in the manuscript that can establish this statement. These statements are functional to the discussion and give the possibility to have further insights on the TE dynamics in soils fertilized with organic amendments.

---

## Editor Comment (EC1) · Karsten Kalbitz (Editor) · 13 Aug 2020

Dear authors, Thank you very much for providing your response to the reviewer's comments. However, I find it quite difficult to get a good overview because it is not easy to distinguish the reviewer's comments (which you did not add to your reply), your discussion and the proposed changes in the manuscript. I suggest to provide for each of the two reviews a separate response file (please just one) with your response (explanation / discussion, changes in the manuscript) which is directly related to the comments of the reviewer. You might use different colors to make the job of the editor easier. Please take into account that your manuscript should not become too much longer than it is. You have to respond to all comments of the reviewer. It is not sufficient to mention that all comments will be considered, e.g. in your response to reviewer 1. There are some

questions in these detailed comments as well and not just the corrections of typos. You should also include all of the comments of reviewer 2 in your response letter. Thank you very much! Karsten Kalbitz

---

## Author Comment (AC4) · 18 Aug 2020

**Referee #1**

R1:The manuscript presented here deals with the model prediction/description of measured data that reflect the Zn, Cu, Cd and Pb dynamics in a long-term field trial amended with different organic amendments. Then, after evaluation, the used model is extrapolated to the future, to evaluate the possible risks of TEs by long-term application of organic amendments on agricultural fields. The advantage of this model is, according to the authors, that it has a restricted amount of input parameters. However, there are several issues with the manuscript in its current state.

1) The model description is not sufficient and scattered over different sections, which makes it difficult to link the different parts in the model. In addition, it should be much more clear which input parameters are needed for each part of the model (i.e., SOM and pH are likely used to calculate Kf's? What are input parameters for the WHAM + which complexes are considered?) + references for ageing parameters are not presented. At least an overview of considered reactions should be presented in the SI, to allow the reader to evaluate the restrictions of the model.

We recognize that the description of IDMM-ag given in the M&M is not sufficient to capture the model details, unless the cited references are checked. This was done partly by purpose to simplify the reading of an already-long manuscript. Therefore, we think that the suggestion given by the Referee to present a model overview is very valuable. In particular, we will review the model description in the M&M and add the following paragraph in the SI.

"

[Figure]

The IDMM simulates a topsoil as a single, fully–mixed soil layer and computes concentrations of metals associated with the soil and in the soil porewater on an annual basis. The soil layer has a defined depth and comprises fine earth material (the soil material itself), coarse matter (stones) and pore space, which is partly filled by water. Annual gains and losses of metal (mol m$^{-2}$ yr$^{-1}$) are computed by a flux balance:

$$\Delta M = M_t + F_{input} + F_{weath} - F_{drain} - F_{crop} \qquad \text{(S1)}$$

where ΔM is the change in metal pool (mol m⁻²), $M_t$ is the metal pool in the soil before calculation (mol m⁻²) and $F_{input}, F_{weath}, F_{drain}, F_{crop}$ are respectively the fluxes of metal into the soil in external input and weathering of metal the coarse fraction, and losses in porewater drainage and due to cropping.

Within the soil, metal is subdivided into a number of forms according to Figure 2A. These are:

1. Metal in porewater, comprising the free metal ion and metal complexed to solution ligands, including dissolved organic matter (DOM);

2. Adsorbed metal, comprising metal reversibly adsorbed to binding sites on the surface of the fine earth material. The sum of the adsorbed metal and the metal in porewater is the 'labile' or 'geochemically active' metal, considered measurable by extraction using a strong ligand or dilute acid (e.g. 0.1M EDTA, 0.43M HNO₃).

3. Two pools of 'aged' metal. This is metal in the fine earth that is considered 'fixed' within the solid phase and only slowly exchangeable with the labile pool. The two pools are characterised by relatively fast and slow exchange kinetics and are termed the 'aged' and 'mineral' pools.

Chemical speciation of the labile metal, including its distribution between the solid phase and porewater, is handled by equilibrium, while exchanges of metal among the labile, weakly aged and strongly aged pools are handled by kinetics.

*Speciation and distribution of the labile metal pool*

The equilibrium speciation and solid–porewater distribution of the labile metal pool is computed annually by a combination of empirical modelling and application of the WHAM/Model VI speciation code (Tipping, 1998). Empirical modelling, following the approach of Groenenberg et al. (2010), is used to derive the relationship between the free metal ion concentration in the porewater and the adsorbed metal concentration, as a function of key soil properties:

$$log\,K_{f.m} = log\{M_{ads}\} - n \cdot log[M^{2+}] = \gamma_1 + \gamma_2 \cdot pH_{pw} + \gamma_3 \cdot log\{SOM\}. \tag{S2}$$

Here $log\,K_{f.m}$ is a Freundlich–type partition coefficient, defined as $log\{M_{ads}\} - n \cdot log[M^{2+}]$ where $\{M_{ads}\}$ is the adsorbed metal concentration (mol g⁻¹), $[M^{2+}]$ is the free metal ion concentration in porewater (mol dm⁻³) and $n$ is a fitted constant in the range zero to unity. The terms $pH_{pw}$ and $\{SOM\}$ are the porewater pH and the soil organic matter content (% w/w) respectively, and the $\gamma$ terms are fitted constants. The model is optimised by fitting to the error in $log\,K_{f.m}$; this drives a set of constants that provide consistent computation of $\{M_{ads}\}$ from $[M^{2+}]$ and vice versa. The constants used are taken from Groenenberg et al. (2010) and are shown in **Table S1**.

**Table S1.** Parameters for adsorbed metal–free ion relationships, adapted from Groenenberg et al., 2010

| Metal | $\gamma_1$ | $\gamma_1$ | $\gamma_1$ | n |
|-------|-----------|-----------|-----------|---|

| | | | | |
|---|---|---|---|---|
| **Cu** | -6.37 | 0.64 | 0.87 | 0.57 |
| **Zn** | -4.67 | 0.46 | 0.84 | 0.84 |
| **Cd** | -5.71 | 0.41 | 0.91 | 0.70 |
| **Pb** | -6.46 | 0.96 | 1.35 | 0.84 |

The relationship between the free metal ion and dissolved metal in the porewater is handled by WHAM/Model VI. Inputs to the model comprise the porewater pH and dissolved concentrations of Na, Mg, Ca, Cl and $SO_4$, and the DOM concentration. These variables are all specified. An initial adjustment is made, whereby either the concentration of Ca, or those of Cl and $SO_4$, are adjusted to achieve charge balance at the specified pH. The speciation of Al is handled using the approach of Tipping (2005) to estimate the activity of $Al^{3+}$ in the porewater on the basis of the pH and DOM concentration. The speciation of FeIII is handled by assuming the porewater $Fe^{3+}$ to be in equilibrium with $Fe(OH)_3$, having a standard solubility constant of 2.5 and a standard enthalpy of -102 kJ mol$^{-1}$.

The equilibrium constants used are shown in Table S2 and Table S3.

Metal binding to DOM is simulated by assuming it to comprise 65% fulvic acid. Each metal has two binding constants, (i) log $K_{MA}$ which is used to derive binding constants for carboxylic and phenolic binding sites, and (ii) $\Delta LK_2$, which is used to generate constants for high affinity binding sites.

**Table S2**. Solution equilibrium parameters for Ca, Al, FeIII and carbonate.

| Equilibrium | log $K^\ominus$ | $\Delta H^\ominus$ (kJ mol$^{-1}$) | Reference |
|---|---|---|---|
| $H^+ + CO_3^{2-} \rightleftharpoons HCO_3^-$ | 10.329 | -14.899 | Nordstrom et al. (1990) |
| $2H^+ + CO_3^{2-} \rightleftharpoons H_2CO_3$ | 16.681 | -24.008 | Nordstrom et al. (1990) |
| $Mg^{2+} + H^+ + CO_3^- \rightleftharpoons MgHCO_3^+$ | 11.4 | -11.59 | Nordstrom et al. (1990) |
| $Mg^{2+} + CO_3^{2-} \rightleftharpoons MgCO_3$ | 2.98 | 11.34 | Nordstrom et al. (1990) |
| $Mg^{2+} + SO_4^{2-} \rightleftharpoons MgSO_4$ | 2.37 | 19.04 | Nordstrom et al. (1990) |
| $Al^{3+} + OH^- \rightleftharpoons AlOH^{2+}$ | 9.01 | -6.44 | May, Helmke, Jackson (1979) |
| $Al^{3+} + 2OH^- \rightleftharpoons Al(OH)_2^+$ | 17.87 | -15.40 | May, Helmke, Jackson (1979) |
| $Al^{3+} + 4OH^- \rightleftharpoons Al(OH)_4^-$ | 33.84 | -45.44 | May, Helmke, Jackson (1979) |
| $Al^{3+} + SO_4^{2-} \rightleftharpoons AlSO_4^+$ | 3.2 | 9.6 | Izatt, Eatough, Christensen, Bartholomew (1969) Sillen and Martell (1971) |
| $Al^{3+} + 2SO_4^{2-} \rightleftharpoons Al(SO_4)^-$ | 5.11 | 13.0 | Izatt, Eatough, Christensen, Bartholomew (1969) Sillen and Martell (1971) |
| $Ca^{2+} + H^+ + CO_3^- \rightleftharpoons CaHCO_3^+$ | 11.44 | -3.64 | Nordstrom et al. (1990) |
| $Ca^{2+} + CO_3^{2-} \rightleftharpoons CaCO_3$ | 3.22 | 14.85 | Nordstrom et al. (1990) |
| $Ca^{2+} + SO_4^{2-} \rightleftharpoons CaSO_4$ | 2.30 | 6.90 | Nordstrom et al. (1990) |
| $Fe^{3+} + OH^- \rightleftharpoons FeOH^{2+}$ | 11.81 | -12.39 | Nordstrom et al. (1990) |

| Equilibrium | log $K^\ominus$ | $\Delta H^\ominus$ (kJ mol$^{-1}$) | Reference |
|---|---|---|---|
| $Fe^{3+} + 2OH^- \rightleftharpoons Fe(OH)_2^+$ | 22.33 | -40.25 | Nordstrom et al. (1990) |
| $Fe^{3+} + 3OH^- \rightleftharpoons Fe(OH)_3$ | 29.44 | -63.97 | Nordstrom et al. (1990) |
| $Fe^{3+} + 4OH^- \rightleftharpoons Fe(OH)_4^-$ | 34.4 | -90.17 | Nordstrom et al. (1990) |
| $Fe^{3+} + SO_4^{2-} \rightleftharpoons FeSO_4^+$ | 4.04 | 16.36 | Nordstrom et al. (1990) |
| $Fe^{3+} + Cl^- \rightleftharpoons FeCl^{2+}$ | 1.48 | 23.43 | Nordstrom et al. (1990) |
| $Fe^{3+} + 2Cl^- \rightleftharpoons FeCl_2^+$ | 2.13 | – | Nordstrom et al. (1990) |

**Table S3.** Solution equilibrium parameters for Cu, Zn, Cd and Pb

| Equilibrium | log $K^\ominus$ | $\Delta H^\ominus$ (kJ mol$^{-1}$) | Reference |
|---|---|---|---|
| $Cu^{2+} + OH^- \rightleftharpoons CuOH^+$ | 6.48 | – | Sunda and Hanson (1979) |
| $Cu^{2+} + 2OH^- \rightleftharpoons Cu(OH)_2$ | 11.78 | – | Sunda and Hanson (1979) |
| $Cu^{2+} + H^+ + CO_3^- \rightleftharpoons CuHCO_3^+$ | 14.62 | – | Mattigod and Sposito (1979) |
| $Cu^{2+} + CO_3^{2-} \rightleftharpoons CuCO_3$ | 6.75 | – | Smith and Martell (1976) |
| $Cu^{2+} + 2CO_3^{2-} \rightleftharpoons Cu(CO_3)_2^{2-}$ | 9.92 | – | Smith and Martell (1976) |
| $Cu^{2+} + SO_4^{2-} \rightleftharpoons CuSO_4$ | 2.36 | 8.8 | Smith and Martell (1976) |
| $Cu^{2+} + Cl^- \rightleftharpoons CuCl^+$ | 0.4 | 6.7 | Smith and Martell (1976) |
| $Zn^{2+} + OH^- \rightleftharpoons ZnOH^+$ | 5.04 | – | Baes and Mesmer (1976) |
| $Zn^{2+} + 2OH^- \rightleftharpoons Zn(OH)_2$ | 11.1 | – | Baes and Mesmer (1976) |
| $Zn^{2+} + H^+ + CO_3^- \rightleftharpoons ZnHCO_3^+$ | 13.12 | – | Mattigod and Sposito (1979) |
| $Zn^{2+} + CO_3^{2-} \rightleftharpoons ZnCO_3$ | 4.76 | – | Mattigod and Sposito (1979) |
| $Zn^{2+} + SO_4^{2-} \rightleftharpoons ZnSO_4$ | 2.38 | 6.3 | Smith and Martell (1976) |
| $Zn^{2+} + Cl^- \rightleftharpoons ZnCl^+$ | 0.4 | 5.4 | Smith and Martell (1976) |
| $Cd^{2+} + OH^- \rightleftharpoons CdOH^+$ | 3.92 | – | Baes and Mesmer (1976) |
| $Cd^{2+} + 2OH^- \rightleftharpoons Cd(OH)_2$ | 7.65 | – | Baes and Mesmer (1976) |
| $Cd^{2+} + H^+ + CO_3^- \rightleftharpoons CdHCO_3^+$ | 11.83 | – | Parkhurst and Appelo (1999) |
| $Cd^{2+} + CO_3^{2-} \rightleftharpoons CdCO_3$ | 4.37 | – | NIST (2003) |
| $Cd^{2+} + 2CO_3^{2-} \rightleftharpoons Cd(CO_3)_2^{2-}$ | 7.26 | – | NIST (2003) |
| $Cd^{2+} + SO_4^{2-} \rightleftharpoons CdSO_4$ | 2.46 | 9.6 | Smith and Martell (1976) |
| $Cd^{2+} + Cl^- \rightleftharpoons CdCl^+$ | 1.98 | 1.3 | Smith and Martell (1976) |
| $Cd^{2+} + 2Cl^- \rightleftharpoons CdCl_2$ | 2.6 | 3.8 | Smith and Martell (1976) |
| $Pb^{2+} + OH^- \rightleftharpoons PbOH^+$ | 6.29 | – | Baes and Mesmer (1976) |
| $Pb^{2+} + 2OH^- \rightleftharpoons Pb(OH)_2$ | 10.88 | – | Baes and Mesmer (1976) |
| $Pb^{2+} + 3OH^- \rightleftharpoons Pb(OH)_3^-$ | 13.94 | – | Baes and Mesmer (1976) |

| Equilibrium | log $K^{\ominus}$ | $\Delta H^{\ominus}$ (kJ mol$^{-1}$) | Reference |
|---|---|---|---|
| $Pb^{2+} + CO_3^{2-} \rightleftharpoons PbCO_3$ | 7.2 | – | Buffle, Chalmers, Masson, Midgley (1988) |
| $Pb^{2+} + 2CO_3^{2-} \rightleftharpoons Pb(CO_3)_2^{2-}$ | 10.5 | – | Buffle, Chalmers, Masson, Midgley (1988) |
| $Pb^{2+} + SO_4^{2-} \rightleftharpoons PbSO_4$ | 2.75 | – | Smith and Martell (1976) |
| $Pb^{2+} + Cl^- \rightleftharpoons PbCl^+$ | 1.59 | 18.4 | Smith and Martell (1976) |
| $Pb^{2+} + 2Cl^- \rightleftharpoons PbCl_2$ | 1.8 | – | Smith and Martell (1976) |

**Table S4.** Constants for metal binding to fulvic acid in WHAM/Model VI. All from Tipping (1998) except where noted.

| Metal | log $K_{MA}$ | $\Delta LK_2$ |
|---|---|---|
| Mg | 1.1 | 0.12 |
| Al | 2.5 | 0.46 |
| Ca | 1.3 | 0 |
| FeIII[a] | 2.6 | 2.20 |
| Cu | 2.1 | 2.34 |
| Zn | 1.6 | 1.28 |
| Cd | 1.6 | 1.48 |
| Pb | 2.2 | 0.93 |

[a] log $K_{MA}$ from Tipping, Rey-Castro, Bryan, Hamilton-Taylor, 2002.

*Exchange between labile, aged and mineral pools*

Metal exchanges between the labile, aged and mineral pools are handled by a first order kinetic schema as introduced by Xu, Lofts and Lu (2016). The schema allows the following transfers of metal among the pools: (i) labile to aged, (ii) aged to labile, (iii) aged to mineral, and (iv) mineral to labile. These transfers are described by the kinetic constants $k_{f,a}$, $k_{f,b}$, $k_{f,m}$ and $k_{b,m}$ respectively. These are summarised in Table S5.

Table S5. Kinetic constants for metals transfers among the labile, aged and mineral pools.

| Metal | $k_{f,a}$ | $k_{f,b}$ | $k_{f,m}$ | $k_{b,m}$ | Reference |
|---|---|---|---|---|---|
| **Cu** | $10^{-2.5+10^{-3.3}e^{pHpw}}$ | $10^{-2.1+10^{-3.5}e^{pHpw}}$ | $10^{-5}$ | | [a] |
| **Zn** | $10^{-4.2+0.26pH_{pw}}$ | $10^{-3.2}$ | $10^{-5}$ | | [a] |
| **Cd** | $10^{-2.9+0.18pH_{pw}}$ | $10^{-1.6}$ | $10^{-5}$ | | [a] |
| **Pb** | $10^{-6.3+0.51pH_{pw}}$ | $10^{-2.6}$ | $10^{-5}$ | | this study |

[a] Xu, Lofts and Lu (2016)

Aging constants for Pb were derived by analysis of an ongoing long term experiment, comprising four UK soils. Soils were spiked with lead and incubated with maintenance of temperature and moisture content (55% of water holding capacity). Samples were periodically taken for quantification of lead spike lability using isotopic dilution. Fitting results are shown in Figure S1.

[Figure]

**Figure S1.** Aging of lead spikes in four UK soils and model fits using the expressions and parameters in Table S5.

**References**

Baes, C.F., Mesmer, A.E. (1976). *The Hydrolysis of Cations*. John Wiley & Sons, New York.

Buffle, J., Chalmers, R.A., Masson, M.R., Midgley, D. (1988). *Complexation reactions in aquatic systems : an analytical approach*. Ellis Horwood, Chichester.

Groenenberg, J. E., Romkens, P. F. A. M., Comans, R. N. J., Luster, J., Pampura, T., Shotbolt, L., de Vries, W. (2010). Transfer functions for solid-solution partitioning of cadmium, copper, nickel, lead and zinc in soils: derivation of relationships for free metal ion activities and validation with independent data. European Journal of Soil Science, 61(1), 58-73. doi:10.1111/j.1365-2389.2009.01201.x

Izatt R. M., Eatough J. J., Christensen J. J. and Bartholomew C. H. (1969) Calorimetrically determined log K, deltaH_0 and deltaS_0 values for the interaction of sulphate ion with several bi- and ter-valent metal ions. J. Chem. Soc. (A) 47-53.

Mattigod, S.V., Sposito, G. (1979). Chemical Modeling of Trace Metal Equilibria in contaminated soil solutions using the computer program, GEOCHEM. In: *Chemical Modeling in Aqueous Systems: Speciation, Sorption, Solubility*. Ed. E. Jenne. ACS Symposium Series 93, American Chemical Society, Washington DC.

May H. M., Helmke P. A., Jackson M. L. (1979) Gibbsite solubility and thermodynamic properties of hydroxy-aluminium ions in aqueous solution at 25°C. Geochimica et Cosmochimica Acta, 43, 861-868.

NIST (2003). *NIST Standard Reference Database 46 – NIST critically selected stability constants of metal complexes: version 7.0*. National Institute of Standards and Technology, US Department of Commerce.

Nordstrom, D. K., Plummer, L. N., Langmuir, D., Busenberg, E., May, H. M., Jones, B. F., Parkhurst, D. L. (1990) Revised chemical equilibrium data for major water-mineral reactions and their limitations In: *Chemical modeling of aqueous systems II*. Eds. D. C. Melchior, R. L.Bassett. ACS Symposium Series 416, p. 398-413, American Chemical Society, Washington DC.

Parkhurst, D.L., Appelo, C.A.J. (1999). *User's guide to PHREEQC (version 2) – a computer program for speciation, batch-reaction, one-dimensional transport, and inverse geochemical calculations*. US Geological Survey, Denver, Colorado.

Sillen L. G. and Martell A. E. (1971) Stability Constants of Metal-Ion Complexes, Suppl. No. 1. Chemical Society, London.

Smith, R.M. and Martell, A.E. (1976). *Critical Stability Constants, Vol. 4: Inorganic Complexes*. Plenum Press, New York.

Sunda, W.G., Hanson, P.J. (1979). Chemical Speciation of Copper in River Water: Effect of Total Copper, pH, Carbonate, and Dissolved Organic Matter. In: *Chemical Modeling in Aqueous Systems: Speciation, Sorption, Solubility*. Ed. E. Jenne. ACS Symposium Series 93, American Chemical Society, Washington DC.

Tipping, E. (1998). Humic Ion-Binding Model VI: An improved description of the interactions of protons and metal ions with humic substances. Aquatic Geochemistry, 4(1), 3-48. doi:Doi 10.1023/A:1009627214459

Tipping, E. (2005). Modelling Al competition for heavy metal binding by dissolved organic matter in soil and surface waters of acid and neutral pH. Geoderma, 127, 293–304. doi:10.1016/j.geoderma.2004.12.003

Tipping, E., Rey-Castro, C., Bryan, S.E., Hamilton-Taylor, J. (2002). Al(III) and Fe(III) binding by humic substances in freshwaters, and implications for trace metal speciation. Geochimica et Cosmochimica Acta, 66(18), 3211-3224. doi: 10.1016/S0016-7037(02)00930-4

Xu, L., Lofts, S., & Lu, Y. (2016). Terrestrial ecosystem health under long-term metal inputs: modeling and risk assessment. Ecosystem Health and Sustainability, 2(5), e01214. doi:10.1002/ehs2.1214

"

R1: 2) The measurements of the TEs lack quality control, the limit of quantification of each element and the relation with the measured data should be presented and, maybe most importantly, the Cd concentrations in the extracts are measured with ICP-OES, what could be troublesome regarding the known interferences during ICP-OES measurements with As in soil extracts.

ICP-OES could give interference of As with Cd measurements, particularly when As is present at medium/high concentrations while Cd is close to the background values (see *McBride. 2011. A comparison of reliability of soil Cd determination by standard spectrometric methods. J Environ Qual. 2011; 40(6): 1863–1869. doi:10.2134/jeq2011.0096*). Unfortunately, we did not measure As concentrations in the field, though we do not expect high enrichment of As in ZOFE soil. However, as a validation of the analytic procedure, we compared the total metal concentration time trends obtained by extraction in aqua regia and analysis of the extracts with ICP-OES with the total metal concentrations obtained from another laboratory in Zurich which analysed the same plots in 2013 and partly in 2014 using extraction in aqua regia and analysis with ICP-MS. In fact, ICP-MS is recognized to have greater sensitivity in Cd determination. We show below the comparison for Cd: measurements from ICP-OES and ICP-MS are in good agreement. This test rules out any possible interference with As. Please note that for determination of total concentrations, the extractions were done with aqua regia and not with 1 M HNO3 as erroneously indicated in the paper.

[Figure]

[Figure]

As to the quality control of the ICP-EOS measurements, it is done on the calibration curve every 10 readings by measuring the TES concentrations in the blanks and in the standard sample at the intermediate concentration of 1 ppm. The limit of quantification for every element is 3 standard deviations of the readings from the blank samples. Therefore, we propose to modify the paragraph in the M&M as follows:

"

The NIL, FYM, SS and COM soils (top 20cm) were sampled from the Agroscope ZOFE soil archive and analysed for total and EDTA-extractable concentrations of Zn, Cu, Pb and Cd. Soils sampled were from years 1972, 1979, 1982, 1991, 1995, 2000, 2003, 2007 and 2011. Before 2011, the samples from the five replicate plots per treatment had been bulked, so that the variability between replicate plots could not be assessed. The archived samples comprised only the 2mm-sieved fraction. To determine total soil TEs concentrations, sample extracts in aqua regia were analysed by means of an inductively coupled plasma optical emission spectrometry (ICP-OES Dv sequential Perkin Elmer Optima 2000). The EDTA-extractable

pools were obtained with the extraction protocol described by Quevauviller (1998) followed by ICP-OES analysis. *Total metal concentration trends obtained with ICP-OES were compared with one-point-in-time measurements from the same plots carried out in other laboratory with ICP-MS, to rule out any interference of As with Cd readings (McBride, 2011). Quality control of the ICP-OES was done on the calibration curve every ten readings by measuring the TES concentrations in the blank samples and in the standard sample at the intermediate concentration of 1 ppm. The limit of quantification for each element was calculated as three standard deviations of the blank readings.* For each metal we took the ratio of the EDTA-extractable concentration and total concentration in the same year to define the metal lability as a measure of the biogeochemically-available fraction at that point in time.

Samples of farmyard manure of years 2011 and 2014, sewage sludge of years 2008 and 2012 and compost of years 2011, 2013 and 2014 were also analysed for total *(by extracting 0.5 g of organic amendment in 10 ml of HNO3)* and EDTA-extractable concentrations of Zn, Cu, Pb and Cd.

"

R1: 3) The implementation of DOC in the model. Because no measurements were available, the DOC is fixed at 7 mg/L for all treatments (as in the beginning of the experiment) and remained constant, despite the application of different organic amendments. I am critical to this approach, due to the important effect of DOC to metal leaching. I would expect increasing (or changing between treatments) DOC concentrations over the years of the different organic amendment applications or at least increased DOC fluxes right after organic amendments that could increase metal leaching.

The Referee raises an important point on the role of the porewater DOC concentration, and we acknowledge that we have not justified the assumptions made regarding its concentrations. Therefore, we propose to introduce the following considerations into the ms.

Data on porewater DOC concentrations from arable soils are scarce and contradictory. To the best of our knowledge, there are no consistent data from long-term experiments. There are a small number of meta-analyses (for example Li et al., 2019; de Troyer et al., 2014), but they are not ideal, because: 1) they do not contain data on soils amended according to all the management approaches taken in ZOFE, so some residual assumptions on DOC would be required; 2) we could not find any data on long-term time trends of field DOC concentrations under arable soils; 3) DOC concentrations obtained from laboratory soil extractions differ from data collected directly from the field using lysimeters, with the latter usually showing lower concentrations; if this is the case, most of the available DOC data are likely to be overestimates of 'true' field concentrations and thus bias the model results. We compared the predictions of the equation suggested by Referee #2 to estimate DOC from pH and SOM (Derivation of Partition Relationships to Calculate Cd, Cu, Ni, Pb, Zn Solubility and Activity in Soil Solutions; Alterra: Wageningen, 2004; p 75) with data from two studies that measured DOC sampled in field using lysimeters:

[Figure]

There is a consistent trend to overestimation of the observed DOC concentration (up to at least two orders of magnitude) and no relationship between observed and predicted DOC concentrations. Therefore, we strongly conclude that the Alterra equation should at best be applied with great care. Application of the equation to the ZOFE plots produced predicted DOC concentrations in the range 60-80 mg/dm3, which we contend, on the basis of the chart above, is highly likely to overestimate DOC and thus not be useful for our purposes.

In conclusion, we think that sources of reliable DOC concentrations for agricultural fields are effectively missing, and so should be identified as key research priorities. Therefore, we suggest that (i) this knowledge gap should be openly confronted and emphasised in the ms; (ii) given this lack of knowledge, the most pragmatic approach is to carry out a systematic sensitivity analysis within a

plausible range of DOC concentrations, incorporating also time trends. We will do a mini-review of field measurements of annual DOC fluxes for temperate grassland sites (before 1949) and arable with/without organic amendments (after 1949) and we will perform simulations with the minimum, maximum and midpoint of our established range of DOC fluxes. To incorporate the time trends in the sensitivity analysis, we propose to apply to each plot a simplified SOC model (for example, a two-pool model like the one in Menichetti et al. (2016), which was previously applied to ZOFE), assuming that DOC correlates with the decomposition fluxes out of these pools. Clearly, this estimation of DOC is affected by the assumptions of C input decomposability when shifting the site management from grassland to arable in 1949; in addition, this approach does not include the effect of soil acidification, which is reported to control the adsorption/desorption of DOC. However, it could be considered a first estimation of DOC time trend.

We have already made some preliminary checks and the main conclusions are the following: increasing the DOC concentration in the control treatment has the effect of increasing the additional flux of metals that we hypothesized to be mineral weathering from the coarse fraction – this is to expected as the assumed DOC and weathering fluxes are not independent since the DOC controls the predicted soil labile concentration at steady state. Also, the impact of increasing five-fold the DOC concentration in the control treatment has a modest effect on the simulated labile concentrations (always <10%). On the other hand, increasing the DOC concentrations in the organic-amended treatments relative to the control treatment has the effect of lowering the metal concentration time-trends in these treatments, thus improving the overall simulation; yet, the effect is also modest.

Finally, the Referee suggests that DOC might peak after amendment application to slowly decrease until next application; this "cycle trend" is likely to happen, however since IDMM-ag has an annual time step, such short term effects are not modelled and an annual average DOC concentration, corresponding to the annual DOC flux, is what the model requires.

References:

Li et al. (2019). Effect of land management practices on the concentration of dissolved organic matter in soil: A meta-analysis". Geoderma 344 (2019) 74–81

De Troyer et al (2014). Factors Controlling the Dissolved Organic Matter Concentration in Pore Waters of Agricultural Soils. Vadose Zone J.

Menichetti et al. (2016.). Parametrization consequences of constraining soil organic matter models by total carbon and radiocarbon using long-term field data. Biogeosciences, 13, 3003-3019.

R1: 4) The artefacts associated with the experimental design of the ZOFE, namely plots touching each other + mixing of plots edges by ploughing. In the model evaluation, it appeared to be critical to introduce lateral mixing. This does not allow to evaluate the model without mixing, which is later used to extrapolate into the future. The lateral spread of TE concentration should be validated in a transect across plots in the ZOFE experiment by some new measurements, to underpin this model approach.

Long-term experiments comprising different treatments are valuable sources of information, but here we showed a potentially unrecognized drawback: plots can be affected by soil mixing with ploughing (plots have a separation space, but eventually it is too narrow for mechanical ploughing). This effect is hardly detectable unless elements present in trace concentrations are taken into account.

Therefore, lateral mixing was introduced to check whether we could fit the data with realistic mixing coefficients. The 4 treatments considered were perfectly adjacent in 3 out of the 5 repetitions (in the other 2 repetitions the compost treatment only was separated from the "block"), so we can argue that the lateral mixing is well represented by the considered treatments and collected data. Clearly, the specific conditions in the long-term experiments cannot be extrapolated to real field conditions, and this is why we did not include the lateral mixing when projecting into the future the TE accumulation. Running another sampling campaign across the whole transect (12 treatments) is not feasible at the moment and, moreover, since some data are missing, it is not clear the real benefits we will get out of such an activity.

R1: In the results, the treatment effects should be evaluated relative to the control data, to evaluate and contribute observed trends to the organic amendments solely, which is the scope of this study.

In the pictures we showed the control treatment and the organic amendment treatments one next to the others because i) the control treatment takes part to the lateral mixing and it's worth showing the relevant simulations ii) absolute concentrations of TEs are useful on their own. However, we could show the metal concentrations of the organic amended treatments relative to the control treatment for the projections into the future.

R1: In addition, both the abstract and introduction lack quantitative data, the English writing could be improved and the final discussion of the results becomes difficult to follow starting from lines 408 to the end.

We will introduce quantitative data as per points below, revise some English mistakes/typos and try to make clearer the ms from line 408.

R1: Further point-by-point comments are presented below.

Abstract

36: abbreviation of model.

Yes, we will give first the full name. thanks.

38: soil plots, different.

OK, we will use different.

Are there more organic amendments than the ones summed up here (particularly is not the best link word here).

Yes, there are other mixed treatments amended with organic amendments and mineral fertilizers.

39-40: link with previous sentence is missing, maybe this sentence can be declined here + don't start sentence with an abbreviation.

Ok, we will get rid of the sentence here.

41: better provide quantitative measure of model performance.

OK, we will report the *r* metric. Thanks.

41: abbreviation ZOFE.

Right, we will replace with full name.

Wouldn't it be interesting to add the range of EDTA-extractable concentrations here in the abstract?

We will.

46: labile = EDTA-extractable? + provide projections, i.e. after XX years, concentrations could increase to YY.

We will use EDTA-extractable here. Yes, we will provide quantitative projections.

Introduction

57-58: this sentence is too vague and is not really necessary here, can be skipped.

OK.

59-64: it would be interesting if you would add the concentration ranges at which the essential and the non-essential TEs become of a concern.

I can provide some mini literature review for the TEs of concern here.

68: an

Yes.

68-71: please rephrase to make the message more clear.

OK. Provisional: "In a European Union wide survey, Ballabio et al. (2018) reported that agricultural soils have one of the highest potential to become enriched in TEs compared with other land uses, and that land cover and management are better predictors of soil Cu concentrations than natural soil formation factors."

71-75: the link with the previous section is not clear. In addition, what do you mean with "limited natural availability of nutrient elements such as P"? Preferably start your sentence with the main message, for example "Organic amendments are considered to be more sustainable then inorganic mineral fertilizers, due to XX and YY".

We mean that P is a limited (not-renewable) natural resource. Provisional rephrase: "Organic amendments are considered to be more sustainable then inorganic mineral fertilizers: for example, current industrial processes for N-fertilizer production are energy-intensive and P-fertilizers are produced from  phosphate rocks  which are naturally limited (Roberts, 2014)".

77: isn't it just the transformation of the organic amendment to SOM that contributes to carbon sequestration as such, not additional C sequestration from atmosphere? If not, please explain more, but only if relevant for this study!

Soil fertility can give a positive feedback to carbon sequestration via vegetation growth.

Actually, line 71-77 could be skipped from this introduction and you could go right at the possible introduction of TEs into soil by organic amendments, to keep the introduction to the point and relevant for this study.

Ok.

77-81: please provide concentration ranges for the TEs in the different organic amendments.

OK. We'll make a mini-review for this.

83: fate? + please rephrase second part of sentence

ok

86: you mean mobility (for solubility)?

Meaning the partition between soil solids and soil solution. We'll rephrase.

87-89: please provide examples or more explanation of importance of TE speciation vs toxicity

Yes, we will.

93: direct reactions?

Binding to soil solids. We will rephrase.

90: I think you can even say that it is not only useful, but even necessary

Ok.

97-98: you already stated this in line 90-91

OK

99-100: Please rephrase: I would not say that mechanistic models are site specific. Indeed, models that predict TE mobility and transfers over time by using as much as possible underlying physical and chemical mechanisms are likely only useful on a limited size scale, due to the high input needed, but they can be used at every site (when data are available), so they are not site specific.

We see your point, therefore we will get rid of "site-specific".

And what about empirical models?

Indeed, IDMM is partly empirical, which alleviates some of the complexity of the mechanistic models. I will expand this concept and merge with the concepts expressed further below the text.

102: what do you mean with behaviour?

Dynamic, we will replace it.

102-106: Ah, here you talk about empirical models. Please merge with text above and try to be more concise.

OK.

112-115: this is vague, could you specify more what variables you are talking about, and which mechanistic level of understanding is wanted.

We will give full model description in the SI, as we answered above.

Has this model been used already (and at what scale)? If yes, please provide the current state-of-the art of the performance on this model. What is the knowledge gap here for this study now?

Thanks for the comment. We will clarify that the model has been used previously at catchment scale, while here we considered a much finer scale.

121: IDMM-ag?

IDMM, I will correct the old naming IDMM-ag.

122: What is "larger scale" here?

Realistic field scale.

123-124: "The hypothesis was that, if the model is successfully applied at field scale with no need of calibration, it might be used at larger scale as well, provided adequate inputs." -> can you test the second part of your hypothesis with this study? If not, please rephrase the hypothesis.

We agree. Provisional version: "The hypothesis was to be able to apply the model to the long-term experiment field with no need of site-specific calibration. Also, we assumed that we could apply the same model to realistic field conditions under same agronomic practices in order to assess their sustainability over time".

125: ZOFE?

We will define the acronym first, thanks.

131-133: please clarify sentence: "large scale", "broad trends", TE concentrations in soil? + rephrase final part of sentence.

OK, we will rephrase, meaning field scale.

M&M

150-152: for TE accumulation, the total applied amendment will likely be important, depending on the data collected on the TE content (per kg of material, per kg of OM,… ?).

We wrote "5 t ha-1 of organic matter every second year…", so we can rephrase it in "5 t OM ha-1 every second year…"

160: 1M HNO3 extractable metals are not total TE concentrations, please just write 1M HNO3 extractable metals. And please also provide more experimental details, L:S ratio, extraction time. In addition, it would be interesting if this 1M HNO3 extracted TEs could be related somehow to "total" element concentrations, measured by aqua regia or XRF or other, more standardized extraction protocols for total soil metal concentration.

As already said above, we made a mistake and we will correct it: the extraction with aqua regia was used for determination of total concentrations.

162-163: please shortly describe, conc. of EDTA, L/S ratio, extraction time.

We wonder if it is really useful to report the protocol described by Quevauviller (1998), as we followed exactly it. Maybe we could spare some space for the results, as the ms is already quite long…

164: try to avoid the use of "we"

ok

164-166: on what is the use of EDTA:1MHNO3 extractable metals as a measure of lability based? Please clarify. Is this already used/tested? If so, please provide references.

See above, we used EDTA/aqua regia extractable metals as a measure of lability.

What about quality control of these measurements, what is the limit of quantification of these methods and how do the measured soil samples relate to this? This could be described already here, or in the results section. In addition, the determination of Cd with ICP-OES in extracts from soil samples

is troublesome due to the interferences with As, even at relative low As soil concentrations. For example see: "A comparison of reliability of soil Cd determination by standard spectrometric methods, M. McBride, JEQ 2011 (40, 1863-1865, doi: 10.2134/jeq2011.0096) and likely many other publications. Did you take this into account? If not, the reliability of the Cd measurements from this study can be severely questioned.

See above for full answer and proposed changes in the ms.

178: preferably write : free and adsorbed TE ions, in contrast to "free TEs"

OK

176-179: please provide an overview of the Freundlich isotherms and TE complexes considered during this study in the supporting information. What are the input parameters for the Freundlich model (i.e. which extraction did you choose to represent the adsorbed fraction, and how does this relate to the adsorbed fraction represented in the initial models of Groenenberg (I think they used 0.43M HNO3 acid extractable metals as "reactive" soil metals). In addition, what other soil properties were measured to calculate the KF by the transfer functions of Groenenberg and how are these soil properties measured? What are the input parameters of the WHAM model and how are they measured?

See above: we will add an extra explanation in the SI

180: please provide the first-order rate constants used for each element and explain wherefrom they are derived (references).

See above: we will add an extra explanation in the SI

184: please provide the start year used in the calculations

Yes, we will, it is 1750.

186-187: please rephrase this sentence, it is not clear what is stated here

Ok, we will. The message is that erosion was neglected as the field was flat.

204-206: please clarify, not clear.

Yes, we will add in the ms that we used the transfer functions reported by Thoni et al. (1996) for Switzerland (Figure S1 in Supporting Information).

207-208: please be consistent in choice of unit

Yes, we will, thanks.

208-211: please provide the data of these fitted mineral weathering fluxes and compare with literature data, if possible.

As said below, we will move this part here from the results.

217: on what are these transfer functions based? Please shortly describe.

Ok.

220: P loading from the manure? You mean addition of P to the fields by manure application? You could quickly give the data here.

Yes, we mean that. Ok we will add the data here.

222: not clear, did you take the X:P ratio's from literature (=1100 data points) or from own measurements (= 2 data points)? Was P also measured in the FYM? That is not been described previously. In addition, how does the measured X:P ratio related to the literature reported?

We will rephrase if not clear. Basically, for Zn and Cu ratios we used the measurements: "The Zn:P and Cu:P ratios were averaged from the values measured in the farmyard manure samples applied in 2011 and 2014 as reported in Table 1". For Pb and Cd we used ratios from literature: "The Pb and Cd inputs with FYM application were also calculated from the P content, using Pb:P and Cd:P ratio values of 0.495 and 0.027, respectively, taken from the work 230 of Menzi and Kessler (2009)"

227: derivation of these factors? One time decrease or decrease linearly with time? Not clear.

These factors were derived from surveys in the cited references. We used a stepwise reduction from 2000 onwards.

231: the detection limit in the caption of table 1 is expressed in mg/L while the concentration data are expressed in mg/kg. Please provide detection limit in mg/kg, to provide a clear idea of the lowest measureable concentration in the FYM. See comment above on the analytics of soils and organic amendments.

Yes, we will change the units. See answer above for detection limits.

232: written like this, Figure S2 is about detection limits, which is not. Please rephrase.

OK

238: tTEs + avoid "we"

OK

245: which peaks? In the soils?

"the peaks measured in the EDTA-extracted concentration trends". We'll add reference to Figure 5 to make it clearer.

246: soil metal concentrations negligible?

It is a typo, we meant sewage sludge metal concentration. Thanks, we will correct it.

Figure S3: in the swiss sludge trend, the Cd fluxes trend is deviating from the other metals. Why?

For Cd, the concentration before 1975 was decreased by a factor of 0.2 because it would give unrealistically/high concentrations. To be added in the explanation.

253-255: Ok at the start of the experiment, but I would expect increasing or changing DOC concentrations over the years of organic amendment applications (or between treatments) or at least DOC fluxes right after organic amendments. I think this approach (constant & low DOC over time) is not a good approach to simulate metal leaching over time in organic amendment treated plots (I assume this constant DOC is then the WHAM input?).

See the answer above.

256: on what is this plausible range based?

We proposed to include a sensitivity analysis on a broad range of DOC values. Still, we think that most DOC measurements in the lab overestimate the field DOC concentrations and that a low DOC value in the NIL treatment is plausible.

257: minor increase for fitted additional input flux? Not clear. + what is minor?

We will change this part with a full sensitivity analysis on DOC (and relevant fitted mineral weathering fluxes)

261: not clear

261: measurements of plant material has not been presented + data for Pb? In addition, changes of plant TE concentrations with changes in labile TE concentration in soils were not considered?

We actually showed estimated crop biomass (from measured product yield) over the course of the long-term experiment in Figure S4. The plant absorption flux in IDMM is calculated, as written, from plant material and fixed plant concentration of TEs.

267-268: It is not clearly explained how SOM and pH affect soild/solution partitioning, aging and speciation, because the input data/model description for the Freundlich, WHAM model and aging model are not well specified.

We will give full model explanation in the SI, see above.

284: pecks?

Typo, it is peaks.

290-291: why?

It is a description of the instrument capabilities…

305-306: Please state how the soil total concentrations are/should be measured in this Swiss Ordinance and comment on own measurement data.

Yes, thanks. Total concentration should be determined by extraction with 2 M HNO3 with a proportion of 1:10 (w/v). We'll make it clear.

311-313: not clear + why 0.1?

We report the answer to Referee #2. We will slightly change the approach and explain it more in details.

We will use a slightly different but more standardized approach to estimate trace element critical limits. This time we will apply: *Lofts et al. (2004). Deriving Soil Critical Limits for Cu, Zn, Cd, and Pb: A Method Based on Free Ion Concentrations. Environ. Sci. Technol. 2004, 38, 3623-3631.* Furthermore, in the future projections we will show the variation of the critical limits over time together with pH and SOM changes. We agree with the Referee that the background concentrations should be subtracted for the calculation of the critical limits; therefore, we propose to subtract the trace element concentrations estimated by the model before 1949 (the start of the experiment).

Results and discussion

324: the trends in the organic amendment plots should be investigated relative to the trends in the control plots, to exclude all other enrichment/losses other than use of organic amendments, which is the core of this study. Then, the statistical analysis should be repeated on these relative data.

As suggested before, considering that the control treatment is involved in the lateral mixing, we propose to keep it separate in the simulations of the long-term data, and to show the relative trends, as suggested by the Referee, only in the future projections.

334-336: compare the measured Cu loss with the literature values + on what is this "expected" Cu leaching based.

OK, we will carry out a mini-literature review on CU losses and we will specify what the leaching is based on (which basically depends on the concentration of dissolved Cu and water leaching from the topsoil).

348-364: same comment, compare treatment effects relative to control.

Same as above.

350-357: you should test correlations between the data to underpin these suggestions.

We will remove most of the text from 322 to 374, and also the suggestions in 350-357.

370: P-overfertilization? Based on what?

We agree. We will rephrase the sentence just saying that P concentration in the sewage sludge was more than twice the P in the farmyard manure and compost.

378-388: this was already (partly) described in the M&M section (and provides answer to above comments), please move this section to the M&M.

Ok.

390: in the figures (also in SI), the ZOFE trend is mentioned. Is this the "Idealized trend"?

Thanks for the comment, yes we improperly used "ZOFE Trend" instead of "Idealized Trend"; we will correct it.

399-401: I have severe doubts of the applicability of the modelling results of TE dynamics to a realistic scale, as the experimental conditions of the field experiment are so specific, i.e., high TE concentration plots "contaminate" low TE concentration plots, so all the treatment effects are obscured by an experimental artefact (i.e. the plowing, the plots being so close to each other…). To be more clear -> the model was not capable to predict the measured concentrations, because the measured concentrations are affected due to the specific design and maintenance of the experimental plots, but such experimental plot are not relevant for real agricultural fields (i.e. narrow soil strips with different amendments that are influenced by lateral mixing), which makes the "fixing" of the model with the lateral mixing not really important for real situations. In addition, due to the fixing of the model by lateral mixing, the true performance of the model cannot be evaluated, and the extrapolation (done in figure 9) to real fields is questionable. However, I understand that this is related to the specific nature of this experimental design and that it is nevertheless worth to investigate the data available, due to the valuable information present from these long-term experiments. However, to verify the overall modelling approach (including lateral mixing and excluding it again to extrapolate the model), I think a simulated transect from figure 6 should be validated by measurements-> i.e. sample along a transect in the ZOFE experiment, measure labile concentrations and remodel for the sampling year.

We already provided an answer to this comment that we report below for convenience.

Long-term experiments comprising different treatments are valuable sources of information, but here we showed a potentially unrecognized drawback: plots can be affected by soil mixing with ploughing (plots have a separation space, but eventually it is too narrow for mechanical ploughing). This effect is hardly detectable unless elements present in trace concentrations are taken into account. Therefore, lateral mixing was introduced to check whether we could fit the data with realistic mixing coefficients. The 4 treatments considered were perfectly adjacent in 3 out of the 5 repetitions (in the other 2 repetitions the compost treatment only was separated from the "block"), so we can argue that the lateral mixing is well represented by the considered treatments and collected data. Clearly, the specific conditions in the long-term experiments cannot be extrapolated to real field conditions, and this is why we did not include the lateral mixing when projecting into the future the TE accumulation. Running another sampling campaign across the whole transect (12 treatments) is not feasible at the moment and, moreover, since some data are missing, it is not clear the real benefits we will get out of such an activity.

402: I guess only the r-value of the Cd is significant? Provide significance of r-values.

Thanks for the suggestion, we will provide the p-value.

408 and further: not clear anymore. Initial measurements underestimated? I've understand that these were fitted?

It was underestimated by a factor of three with the "Swiss Sludge Trend" as opposed to the "Idealized Trend". We will be clearer in the sentence.

471-472: but the EDTA-extractable concentrations were measured? Couldn't this provide information of the "lability" of the TE input by the organic amendments.

We measured the "lability" of the TE input by the organic amendments only in the few available samples: in 2011 and 2014 for farmyard manure, in 2008 and 2012 for sewage sludge and in 2011, 2013, 2014 for compost (see Figure 4). The TE lability in the sewage sludge was highly variable; we found that the organic fraction of the sewage sludge was variable as well. Unfortunately, we didn't have a full time trend to fit in the model, therefore we considered the lability data as qualitative. Furthermore, the TE lability measured in the organic amendments did not always have a good match with the lability measured in the soil (for example, for Zn).

---

## Author Comment (AC5) · 18 Aug 2020

**Referee #2**

R2: The manuscript presented here deals with the model prediction/description of measured data that reflect the Zn, Cu, Cd and Pb dynamics in a long-term field trial amended with different organic amendments. Then, after evaluation, the used model is extrapolated to the future, to evaluate the possible risks of TEs by long-term application of organic amendments on agricultural fields. The advantage of this model is, according to the authors, that it has a restricted amount of input parameters. However, there are several issues with the manuscript in its current state.

We thank the referee for acknowledging the interest in the ms.

R2: I have however a range of comments that to my point of view should be addressed before addressing this MS for publication in SOIL.

Major comments My first major comment is related to the general approach followed and to what is to my point of view the main conclusion of the paper. The author concluded (lines 509-511) that "the IDDM-ag model provided an adequate description of the measured EDTA-extractable concentration trends: : :". Looking at the figure 5, this is not so obvious. There are indeed several situations for which there is a clear discrepancy between experimental data and modelling (e.g. Zn-SS, Pb-SS, Cd-COM) and also other situations where the Swiss (e.g. Zn-FYM, Pb-FYM) or the ZOFE (e.g. Cu-SS) was alternatively the hypothesis which enables the model to have the best fit. In addition, these fits are based on the modelling approach considering lateral mixing. If the principle of lateral mixing is explained and if the uncertainty related to such a computation is discussed in the MS, there is not any validation of such a computation based on experimental data. In addition to that, there are a lot of uncertainty on some major flux of trace elements for a significant part of the field experiment history, particularly on trace elements added by organic residues. To consider whether the fits obtained were adequate or not, uncertainty in model parameters and input data should be considered and compared to the uncertainty in experimental data.

We acknowledge the fact that there is great uncertainty in the input data, as well as on the mechanisms governing TEs accumulation in the soil; decoupling and/or quantifying these uncertainties is out of the scope of the ms. Yet, the approach followed to quantify the inputs was able to reproduce the observed trends with lateral mixing and under the "Idealized Trend" assumption. Said this, we recognize that stating that simulations were "adequate" would require additional considerations, therefore we won't use this term in the ms.

For what concerns lateral mixing, we report the answer to Referee#1.

Long-term experiments comprising different treatments are valuable sources of information, but here we showed a potentially unrecognized drawback: plots can be affected by soil mixing with ploughing (plots have a separation space, but eventually it is too narrow for mechanical ploughing). This effect is hardly detectable unless elements present in trace concentrations are taken into account. Therefore, lateral mixing was introduced to check whether we could fit the data with realistic mixing coefficients. The 4 treatments considered were perfectly adjacent in 3 out of the 5 repetitions (in the other 2 repetitions the compost treatment only was separated from the "block"), so we can argue that the lateral mixing is well represented by the considered treatments and collected data. Clearly, the specific conditions in the long-term experiments cannot be extrapolated to real field conditions, and this is why we did not include the lateral mixing when projecting into the future the TE accumulation. Running another sampling campaign across the whole transect (12 treatments) is not feasible at the moment and, moreover, since some data are missing, it is not clear the real benefits we will get out of such an activity.

Moreover, considering the question about the adequacy of the fits of EDTA-extractable concentration, it should be to my point of view necessary to show as the first step the fits obtained for total trace element concentration trends in soil. The idea is that if the total trace element trends are not adequately simulated, how the EDTA-extractable trend could be? The simulation of total trace element concentration trends seem to me even more necessary as the accumulation trend expected is not visible for most trace element and organic fertilisation modalities. In particular, total Cu concentration shows a strong decreasing trend that was attributed to the past application of Cu fungicide, then followed by a sharp removal of Cu from the top-soil layer. No convincing explanation is given for this as the authors said that they do not have information about Cu-fungicide applications and assumed that soil ploughing and bioturbation explained the dilution of Cu concentration in soil without any simulation to support these strong assumptions. Without any other explanation, it seems that the Cu dataset is strongly biased and should, to my point of view, be removed from the MS.

The IDMM is mainly designed to simulate labile metal concentrations, but we agree that it would be informative to show the total metal concentration simulations as well. Therefore, we will include them, but we propose not to disregard the Cu data. Indeed, we do recognize that the hypothesis of fungicide application and bioturbation/soil removal is not conclusive, but we should also consider that: i) the Cu labile concentration trends are plausible and a valuable source of information; ii) we should give evidence in the literature of those data which could not be explained satisfactorily, for future research; iii) there is no obvious reason why Cu total concentrations should be biased, while the total concentrations of the other metals should not be so. Therefore, we propose to show the total concentration simulations for Zn, Cu, Cd, Pb, leaving open the question of the large decrease of Cu, which the model will not simulate.

R2: My second major comment is related to the second major conclusion of the paper suggesting that Cu and Zn contamination in soil can be harmful to soil organisms. To my understanding, this conclusion is based on the methodology described lines 309-313. It is however really unclear how the related computation of critical limits was effectively achieved. I looked at the cited paper of Lofts et al. 2005, from which I supposed that the free ion approach was based on EDTA-extractable concentration, pH and SOM data. If I am right, I notably wonder how the natural background concentration of trace elements in the soil was considered as regard to the fact that this specific issue is addressed by Lofts et al. (2005). Also, this methodology was tested on two dataset from UK and North America. It is thus not obvious that the methodology is relevant for the specific case and consequently the specific application of the IDDM-ag model studied here.

We propose to use a slightly different but more standardized approach to estimate trace element critical limits. This time we will apply: *Lofts et al. (2004). Deriving Soil Critical Limits for Cu, Zn, Cd, and Pb: A Method Based on Free Ion Concentrations. Environ. Sci. Technol. 2004, 38, 3623-3631.* Furthermore, in the future projections we will show the variation of the critical limits over time together with pH and SOM changes. We agree with the Referee that the background concentrations should be subtracted for the calculation of the critical limits; therefore, we propose to subtract the trace element concentrations estimated by the model before 1949 (the start of the experiment).

R2: Additional comments Lines 67-68 and 77-82: Sewage sludge is introduced differently from other organic residues (FYM and COM), notably because of the higher trace element concentration found in SS compared to FYM and COM. However, this is because the FYM and COM had relatively low trace element concentrations. For instance, I suppose that FYM is a cow manure. If a pig or poultry manure

had been chosen, the concentration of several trace elements (Cu and Zn more particularly) would have been much higher and likely comparable to the concentrations observed in SS.

The organic amendments are applied at different frequencies and quantities in order to introduce the same (estimated) amount of organic matter, so that the comparison is done on the same basis. Since the TEs concentrations had not been assessed before this work, they did not contribute to the decision of applying different rates of organic amendment. Said this, the Referee is right that the FYM is from cow and the comment is valid that FYM from different sources could have much higher TEs concentrations. We will rephrase the sentence.

Line 199: To what refer the metal input? To the pool of total or available trace element?

In the model they are added to the labile pool.

Lines 199-211: It is not clear what is considered behind "geogenic input". It is also very surprising to use data on the weathering rate of deppe layers of peat bogs to estimate the weathering rate in the present field experiment where the soil is clearly not a peat bog.

By geogenic deposition we mean the natural concentration of TEs in the atmosphere, i.e. due to eruptions from volcanoes, which give rise to TEs deposition well before anthropogenic activities became prevalent. In fact, this deposition would be detectable even at "pristine conditions". This is why we used data from deep peat bog layers to estimate this natural deposition of TEs (please note that we did not use peat bog data to estimate mineral weathering!).

Doc concentration was fixed at 7 mg C/L. The authors further argued that Doc variation between 7 and 12 mgC/L does not impact the leaching of TE. However, considering the large variation in SOM and pH in the different fertilization modalities, I am surprised that a larger range of Doc concentration was not expected. Several authors (e.g. Araujo et al. 2019, https://doi.org/10.1016/j.envpol.2018.12.070; Cambier et al. 2014, https://doi.org/10.1016/j.scitotenv.2014.06.105; Laurent et al. (2020, https://doi.org/10.1016/j.scitotenv.2019.135927) showed drastic change in Doc concentration in soil amended with organic residues. A way to estimate the initial Doc concentration and its likely evolution over time could have been to use the empirical multi-linear regression suggested by Romkens et al. 2004 (Derivation of Partition Relationships to Calculate Cd, Cu, Ni, Pb, Zn Solubility and Activity in Soil Solutions; Alterra: Wageningen, 2004; p 75).

Please, see the answer to Referee 1 reported below.

The Referee raises an important point on the role of the porewater DOC concentration, and we acknowledge that we have not justified the assumptions made regarding its concentrations. Therefore, we propose to introduce the following considerations into the ms.

Data on porewater DOC concentrations from arable soils are scarce and contradictory. To the best of our knowledge, there are no consistent data from long-term experiments. There are a small number of meta-analyses (for example Li et al., 2019; de Troyer et al., 2014), but they are not ideal, because: 1) they do not contain data on soils amended according to all the management approaches taken in ZOFE, so some residual assumptions on DOC would be required; 2) we could not find any data on long-term time trends of field DOC concentrations under arable soils; 3) DOC concentrations obtained from laboratory soil extractions differ from data collected directly from the field using lysimeters, with the

latter usually showing lower concentrations; if this is the case, most of the available DOC data are likely to be overestimates of 'true' field concentrations and thus bias the model results. We compared the predictions of the equation suggested by Referee #2 to estimate DOC from pH and SOM (Derivation of Partition Relationships to Calculate Cd, Cu, Ni, Pb, Zn Solubility and Activity in Soil Solutions; Alterra: Wageningen, 2004; p 75) with data from two studies that measured DOC sampled in field using lysimeters:

[Figure]

There is a consistent trend to overestimation of the observed DOC concentration (up to at least two orders of magnitude) and no relationship between observed and predicted DOC concentrations. Therefore, we strongly conclude that the Alterra equation should at best be applied with great care. Application of the equation to the ZOFE plots produced predicted DOC concentrations in the range 60-80 mg/dm3, which we contend, on the basis of the chart above, is highly likely to overestimate DOC and thus not be useful for our purposes.

In conclusion, we think that sources of reliable DOC concentrations for agricultural fields are effectively missing, and so should be identified as key research priorities. Therefore, we suggest that (i) this knowledge gap should be openly confronted and emphasised in the ms; (ii) given this lack of knowledge, the most pragmatic approach is to carry out a systematic sensitivity analysis within a plausible range of DOC concentrations, incorporating also time trends. We will do a mini-review of field measurements of annual DOC fluxes for temperate grassland sites (before 1949) and arable with/without organic amendments (after 1949) and we will perform simulations with the minimum, maximum and midpoint of our established range of DOC fluxes. To incorporate the time trends in the sensitivity analysis, we propose to apply to each plot a simplified SOC model (for example, a two-pool model like the one in Menichetti et al. (2016), which was previously applied to ZOFE), assuming that DOC correlates with the decomposition fluxes out of these pools. Clearly, this estimation of DOC is

affected by the assumptions of C input decomposability when shifting the site management from grassland to arable in 1949; in addition, this approach does not include the effect of soil acidification, which is reported to control the adsorption/desorption of DOC. However, it could be considered a first estimation of DOC time trend.

We have already made some preliminary checks and the main conclusions are the following: increasing the DOC concentration in the control treatment has the effect of increasing the additional flux of metals that we hypothesized to be mineral weathering from the coarse fraction – this is to expected as the assumed DOC and weathering fluxes are not independent since the DOC controls the predicted soil labile concentration at steady state. Also, the impact of increasing five-fold the DOC concentration in the control treatment has a modest effect on the simulated labile concentrations (always <10%). On the other hand, increasing the DOC concentrations in the organic-amended treatments relative to the control treatment has the effect of lowering the metal concentration time-trends in these treatments, thus improving the overall simulation; yet, the effect is also modest.

Finally, the Referee suggests that DOC might peak after amendment application to slowly decrease until next application; this "cycle trend" is likely to happen, however since IDMM-ag has an annual time step, such short term effects are not modelled and an annual average DOC concentration, corresponding to the annual DOC flux, is what the model requires.

References:

Li et al. (2019). Effect of land management practices on the concentration of dissolved organic matter in soil: A meta-analysis". Geoderma 344 (2019) 74–81

De Troyer et al (2014). Factors Controlling the Dissolved Organic Matter Concentration in Pore Waters of Agricultural Soils. Vadose Zone J.

Menichetti et al. (2016.). Parametrization consequences of constraining soil organic matter models by total carbon and radiocarbon using long-term field data. Biogeosciences, 13, 3003-3019.

Line 303: The choice to fix pH and SOM in soil at the value found in 2014 for predictive modelling is really disputable, when considering how these two parameters are strongly impacted by the long-term applications of organic residues and particularly in the context the field experiment studied as showed in figure S4. This point should at least be discussed.

The Referee is right that even after >65 yr of soil management, none of the plots here considered have reached a new equilibrium condition: pH and SOC are still decreasing, sometimes at lower rates than initially. Therefore, keeping fixed pH and SOC can be a crude assumption. We propose to apply a SOC model to predict the future SOC (and DOC) changes over time (see answer just above). For pH, since its value will depend upon a number of factors, not least the speciation and cycling of added N, we propose to use simple extrapolation of the observed trends.

Lines 324-338 and 348-364: These two paragraphs are really too descriptive and speculative. As related to my first main comment, the simulation of total trace element concentration trends seem a prerequisite to assess the adequacy of the model used. But, as far trace element availability in soil is concerned, some (usually found in the literature, but nevertheless strong) hypotheses on the soil

parameters explaining the change over time in the EDTA-extractable concentration of trace elements. These hypotheses should be checked, for instance by looking for multi-linear regression between trace element EDTA-extractable concentration or lability and some important parameters such as the input of trace metal in soils, total trace metal concentration in soil, SOM and pH in soil.

We will delete the description of the measured data and give space to the additional work that these comments have raised (i.e. DOC sensitivity analysis). Though it would be interesting to apply a multi-linear regression to explain the measured data, we feel it would be out of the scope of this ms, which is focused on model application and future predictions of TEs bioavailable concentrations.

Lines 425-445: Basically, the consideration of lateral mixing is interesting. However, the comparison of simulations with and without lateral mixing should be showed clearly (at least in supporting information) to support the conclusion that accounting for lateral mixing is important.

Yes, thanks. We will show the comparison with/without lateral mixing in the SI.

Section 3.4: It is really unclear to me what is the added value of the FTIR and XRD datasets. To my point of view, these datasets should be removed.

The spectroscopic analysis is useful for two important perspectives: i) confirming the general reduction of organic matter in soil for long term treatments in all samples; ii) confirming the importance to know the nature of organic material in terms of high affinity for TE's and its possible consequence in affecting the model. The X-Ray Diffraction analysis on soil samples give the possibility to establish that the organic treatments have not introduced exogenous mineral material, especially, in the case of the sewage sludge application. Differently, there are no other data in the manuscript that can establish this statement. These statements are functional to the discussion and give the possibility to have further insights on the TE dynamics in soils fertilized with organic amendments.

We propose to keep these sections, eventually shortening them.

---

## Author Response (AR2)

Referee 1 expressed a positive evaluation for ms acceptance and a few typos/suggestions that we have implemented. Regarding Referee 2 we provide below the following comments.

Total vs labile concentrations. IDMM is a model mainly devoted to labile metal simulations. In fact, total concentrations add up the contributions of labile and (prevalent) non-labile metal concentrations. From a mechanistic point of view, the dynamics of labile and non-labile metal concentrations are very different; IDMM not only treats the two pools in different ways, but relevant parameters are constrained with different dataset and experiments. The labile pool processes are mainly short-term equilibria (and so are the relevant datasets), while the non-labile processes are long-term (irreversible?) processes. We provided evidence of such processes in Figure 2. As the understanding of the two kinds of processes have different levels of confidence, models that use them will reflect the same knowledge gap. This is the reason why we presented the total metal concentrations in the Supporting Information. For example, for Pb we used fresh kinetic constants from a long-term lab experiment: NIL and FYM observations were well estimated while SS and COM were overestimated. Zn was well reproduced, except for underestimation in SS. On the other hand, Cd was overestimated in all the treatments. Surely, Cu is a different story: the total concentrations showed a strong reduction over time that we could not explain. Yet, we decided to keep Cu by offering the motivation that labile metal concentrations have reasonable (and well reproduced) trends. Therefore:

- Models, like knowledge, need time and trials to accumulate confidence and robustness, this is the basics of research. What we presented is a quite challenging simulation of labile concentrations (which gave promising results) and less satisfactory simulations of total metal concentrations, thought total metal simulations were better in some cases and worse in others, but, except for Cu, the trends were overall matched. What we present here could be the basis for future studies and developments.
- 2) The argument that if total metal concentrations are not perfectly reproduced, then the labile metal simulations should be dismissed has no process-driven foundation. The mechanisms behind labile and non-labile metal dynamics are related but different.
- 3) We did not enter the question of whether total or labile concentrations should be used for risk assessment, nor should the Referee do so. Total and labile pools have different meanings; we concentrated on the labile pool as IDMM is fundamentally a labile-pool dynamic model.

ZOFE long-term trial. The argument provided by the Referee is that the dataset from ZOFE long-term trial should be dismissed as it conflicts with the bulk of the literature. First, we don't think so as the metal total concentrations generally increased or reached a plateau in the long-term. This was not true for copper (and we provided extensive comments that we could not justify such a loss) and in the sewage sludge treatment, for which we attributed such a behaviour to soil lateral mixing. In fact, we think that it is hard, if not impossible, to reply to the opinion that a dataset should be dismissed if it does not conform to certain expectations. Unless qualitative and quantitative justifications are provided, data should not be dismissed on the basis of unjustified feelings.

Regarding soil lateral mixing, this effect was already reported at Rothamsted long-term experiment, and there the datasets have been used for publication. While long-term experiments have unique long-term records for metal dynamics, they might come with the disadvantage of having small plots. While the soil lateral mixing effect is not significant for soil organic matter simulations, it is for trace metals. The proposal of the referee is to carry out the simulations for another long-term experiment, which should serve for validation. We argument that this is a totally different work and that there is no guarantee that the other work will have unambiguous data to validate soil lateral mixing!

FTIR and XRD. These two techniques were primarily used to exclude the contribution of OM/mineral composition variation to the decreasing trends observed in the sewage sludge amended plots. Therefore, the Referee is wrong in saying that FTIR was used to quantify SOC loss (for which data are available and used in the simulations) or make a comparison between treatments. The Referee asks whether we knew that sewage sludge, and relevant minerals, were added in substantial quantity to modify the mineral phase composition. The fact is that we didn't know, but now that we have the XRD dataset we can exclude it. On the other hand, the FTIR revealed a change in the SOM composition. How relevant it was for metal availability is difficult to quantify, but we notice that Pb availability decreased over time in contrast with the other metals. This might have an effect in TEs modelling, because the lability of the incoming metals could be tuned rather than being kept constant (this was already noticed by Bergkvist and Jarvis (2004). We suggest that, if these information are not provided, someone might ask whether the amendment itself contributed to the observed trends.

Finally, as an answer to the Editor's point, we have added a table with the limits of quantification and we have shortened and clarified the part on FTIR/XRD analysis.